# PROCEDURAL MISTAKE DETECTION VIA ACTION EFFECT MODELING

**Wenliang Guo, Yujiang Pu, Yu Kong**
Michigan State University, USA
{guowenli,puyujian,yukong}@msu.edu

## ABSTRACT

Mistake detection in procedural tasks is essential for building intelligent systems that support learning and task execution. Existing approaches primarily analyze how an action is performed, while overlooking what it produces, i.e., the **action effect**. Yet many errors manifest not in the execution itself but in the resulting outcome, such as an unintended object state or incorrect spatial arrangement. To address this gap, we propose Action Effect Modeling (AEM), a unified framework that jointly captures action execution and its outcomes through a probabilistic formulation. AEM first identifies the outcome of an action by selecting the most informative effect frame based on semantic relevance and visual quality. It then extracts complementary cues from visual grounding and symbolic scene graphs, aligning them in a shared latent space to form robust effect-aware representations. To detect mistakes, we further design a prompt-based detector that incorporates task-specific prompts and aligns each action segment with its intended execution semantics. Our approach achieves state-of-the-art performance on the EgoPER and CaptainCook4D benchmarks under the challenging one-class classification (OCC) setting. These results demonstrate that modeling both execution and outcome yields more reliable mistake detection, and highlight the potential of effect-aware representations to benefit a broader range of downstream applications. [1]

## 1 INTRODUCTION

Mistakes are inevitable in procedural tasks, from cooking and assembly to medical procedures. While humans refine their skills through learning and experience, they often overlook subtle execution details that lead to unintended outcomes. For instance, a person might follow what seems to be the correct cutting motion, yet still produce irregular cucumber slices due to a slight misalignment in technique. These errors may not be immediately apparent from the action itself but become evident in the final result. To better support users in performing tasks, intelligent systems must therefore assess not only how actions are performed, but also what those actions ultimately produce.

Existing mistake detection approaches focus primarily on modeling the execution process, assessing correctness by analyzing motion patterns and action sequences. Previous works have explored such methods in structured tasks, such as assembly (Sener et al., 2022) and cooking (Wang et al., 2023; Peddi et al., 2023), often relying on frame-level classification or multimodal features. More recently, Lee et al. (2024) introduced prototype learning to evaluate execution similarity across instances, while Huang et al. (2025a) utilized task graphs to predict all possible next actions and compared them with actual executions in the latent space. However, a shared limitation of these methods is the assumption that mistakes can be identified solely from the execution process, without verifying whether the final outcome aligns with the intended result.

In real-world scenarios, the execution may appear correct, yet minor deviations can lead to flawed outcomes. As illustrated in Figure 1a, an imprecise stirring position might resemble a correct motion, while the resulting spillage reveals the error. In Figure 1b, subtle differences in slicing technique may produce irregularly shaped slices. These examples underscore a critical limitation: mistake detection should account for not only the execution process but also the resulting action effect.

---

[1]Project page: https://wenliangguo.github.io/Mistake_Detection

Activity: Make Butter Corn Cup    Action: Stir Mixture in the Bowl

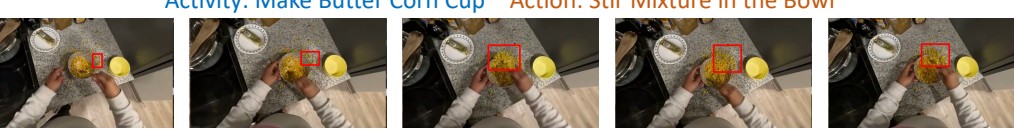

(a) Positional Mistake, i.e., spill out mixture onto table.

Activity: Make Cucumber Raita    Action: Cut Cucumber to Pieces

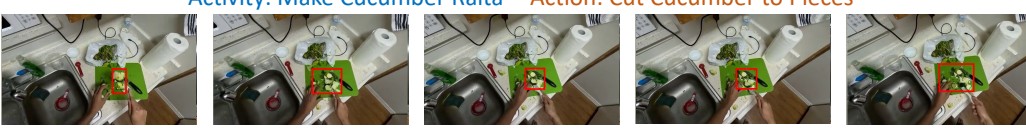

(b) State mistake, i.e., cucumber is cut into unexpected shape.

Figure 1: Examples of two types of mistakes: object positional and state mistakes, in cooking scenario from CaptainCook4D dataset (Peddi et al., 2023).

We address this challenge by casting mistake detection as a marginalization problem over latent variables including potential action effects. Concretely, the probability of a mistake is modeled as a joint function of both the execution of an action and its outcome. This probabilistic view naturally decomposes the task into three subproblems: identifying the action effect, interpreting the effect, and determining whether it reveals a mistake. Based on this formulation, we propose **Action Effect Modeling (AEM)**, a unified framework that enriches action representations with effect-aware features. AEM first selects an effect frame that best reflects the result based on both semantic relevance and visual quality. The action effect is then modeled from two complementary perspectives: *object states*, which capture visual attributes such as shape and size changes, and *spatial relationships*, which describe how objects are arranged relative to one another.

To learn robust and interpretable effect-aware representations, AEM leverages multimodal supervision from vision-language models (Hurst et al., 2024) through dual branches. The visual branch encodes grounded object features and positions, while a textual branch constructs symbolic scene graphs to extract attributes and relational descriptions. During training, these supervision signals are aligned in a shared latent space via contrastive objectives, allowing the model to distill structured effect knowledge into a compact representation. At test time, the learned effect representation is fused with the action segment features, providing enriched context for downstream mistake detection.

While action effects provide valuable cues, effective mistake detection also requires modeling the execution itself. To this end, we develop a prompt-based detector that evaluates whether an action segment aligns with its execution process within the task context. Rather than relying on frame-level classifiers, our detector encodes each action using a learnable representation and aligns it with a task-specific textual prompt in a contrastive manner. By modeling both the temporal dynamics and the resulting effect of an action, the model captures rich semantic cues to distinguish subtle execution errors as well as outcome discrepancies. Our approach is evaluated on two challenging egocentric datasets, EgoPER (Lee et al., 2024) and CaptainCook4D (Peddi et al., 2023), achieving state-of-the-art performance. Extensive ablations further demonstrate the necessity and effectiveness of action effect modeling in improving mistake detection.

In summary, our contributions are threefold:

- We formulate mistake detection as a probabilistic marginalization over latent action effects, decomposing the task into effect frame sampling, effect modeling, and mistake classification.
- We propose Action Effect Modeling to learn effect-aware action representations by capturing object state and spatial relationships using complementary visual and symbolic cues.
- We develop a prompt-based detector that aligns action segments with task-specific textual prompts, enabling effective detection of both execution errors and outcome errors in context.

## 2    RELATED WORK

**Mistake Detection** consists of two main branches: online detection (Jang et al., 2019; Ding et al., 2023; Narasimhan et al., 2023; Ashutosh et al., 2024; Flaborea et al., 2024; Plini et al., 2024; Soran

et al., 2015; Jang et al., 2023) and offline detection (Sener et al., 2022; Wang et al., 2023; Haneji et al., 2024; Peddi et al., 2023; Schoonbeek et al., 2024; Ghoddoosian et al., 2023; Lee et al., 2024; Mazzamuto et al., 2024; Storks et al., 2024; Huang et al., 2025a). Online mistake detection focuses on detecting sequential mistakes in real time, while offline detection aims to recognize incorrect action execution. This work primarily addresses offline mistake detection.

Zooming into offline detection, early approaches relied on training binary classifiers with visual or multimodal input (e.g., text, audio, depth) to learn implicit action representations, without explicitly modeling the underlying action dynamics (Sener et al., 2022; Haneji et al., 2024; Peddi et al., 2023). More recently, Lee et al. (2024) introduced a prototype-based method that explicitly models actions by clustering features during training to learn representative action embeddings. Huang et al. (2025a) utilized task graphs to predict possible next steps and reconstructed action feature in the latent space. Both methods realized error detection through comparison of similarity between the predicted and input action features at inference time.

While these methods achieve strong performance, they predominantly focus on modeling the execution process. However, the inherent variability of human behavior poses a significant challenge for detecting fine-grained mistakes solely from execution dynamics. This limitation motivates us to go beyond modeling the action itself by incorporating action effects as complementary signals, thereby enhancing the robustness and reliability of mistake detection.

**Video Anomaly Detection** (Wu et al., 2019; Gong et al., 2019; Pu & Wu, 2022; Wu et al., 2022; Al-Lahham et al., 2024; Pu et al., 2024) is closely related to mistake detection, where models are typically trained on normal videos and identify anomalies as statistical deviations, without requiring prior knowledge of specific abnormal events. However, mistake detection in procedural videos is inherently goal-oriented, as it requires an understanding of structured workflows to assess whether an action is executed correctly. In addition, VAD is predominantly used in surveillance systems, where the camera view remains fixed and the dynamics of the scene is relatively stable. In contrast, procedural videos, especially in egocentric views, involve more fine-grained human activities, explicit scene transitions, and dynamic human motion, making action-effect modeling more challenging but crucial for building life assistive system.

## 3 METHODOLOGY

### 3.1 PROBLEM FORMULATION

Given a procedural video corresponding to a predefined task, our goal is to determine whether each segmented action is executed correctly or contains a mistake. We operate on non-overlapping segments, where each segment is denoted as a tuple $s = (t_s, t_e, a, y)$, with $t_s$ and $t_e$ denoting the start and end timestamps, $a$ the action label, and $y \in \{0, 1\}$ the binary mistake label ($y = 1$ indicates a mistake). Following the one-class classification (OCC) setting (Lee et al., 2024; Huang et al., 2025a), the training set includes only normal actions ($y = 0$), while the test set contains both correct and erroneous segments.

Formally, the objective is to estimate the probability of a mistake given a visual representation of the segment, expressed as $P(\hat{y} \mid \mathbf{X})$, where $\mathbf{X}$ denotes the encoded segment feature and $\hat{y}$ is the predicted mistake label. Unlike prior methods that classify segments solely based on execution dynamics, we argue that procedural correctness depends on both the execution process and its outcome. This motivates a probabilistic formulation in which mistake detection is expressed as a marginalization over effect frames and the latent effect descriptors:

$$\sum_{i=1}^{K} \sum_{f_e} \underbrace{P(\hat{y} \mid \mathbf{X}, e_i)}_{\text{mistake detection}} \cdot \underbrace{P(e_i \mid f_e, \mathbf{X})}_{\text{effect-aware learning}} \cdot \underbrace{P(f_e \mid \mathbf{X})}_{\text{frame sampling}}, \tag{1}$$

where $f_e$ denotes the discrete effect frame sampled from the video, and $\mathbf{e}_i$ represents the $i$-th possible descriptor of the action effect extracted from $f_e$. $K$ is the total number of effect descriptors, which in our case includes two complementary perspectives: object states and spatial relationships, as they reveal common mistakes in procedural activities. Here we assume that once the segment feature $\mathbf{X}$ and the effect descriptor $e_i$ are obtained, the specific effect frame $f_e$ does not provide additional

information for mistake prediction, i.e., $\hat{y} \perp f_e \mid (\mathbf{X}, e_i)$, and $P(\hat{y} \mid \mathbf{X}, e_i, f_e) = P(\hat{y} \mid \mathbf{X}, e_i)$. Intuitively, Eq. 1 indicates that we consider every possible effect frame and each of its associated action-effect descriptors, estimate the likelihood of each $(f_e, e_i)$ combination, evaluate whether it suggests a mistake, and then aggregate these weighted probabilities to obtain the final prediction.

This formulation provides a structured view of the mistake detection pipeline: frame sampling, effect-aware learning, and mistake classification, bridging the gap between how an action is performed and what outcomes it produces. Next, we will build a framework following this formulation.

## 3.2 FRAMEWORK OVERVIEW

Figure 2 illustrates the overall framework. We first adopt an action segmentation backbone to obtain frame-wise features and non-overlapping action segments. The frame features are concatenated along the temporal dimension based on segment boundaries. The resulting segment features $\mathbf{X}$ are fed into the Action Effect Modeling (AEM) module which (i) performs effect frame sampling $P(f_e \mid \mathbf{X})$ to identify the frame most indicative of the outcome, (ii) extracts multimodal effect knowledge $P(\mathbf{e} \mid f_e, \mathbf{X})$ including object states and spatial relationships, and (iii) learns an effect-aware representation aligned across visual and symbolic cues. Finally, the enriched segment features are passed into the mistake detection module, which models $P(\hat{y} \mid \mathbf{X}, \mathbf{e})$ using a prompt-based detector. In the following, we detail three core components: action segmentation, effect modeling and mistake detection.

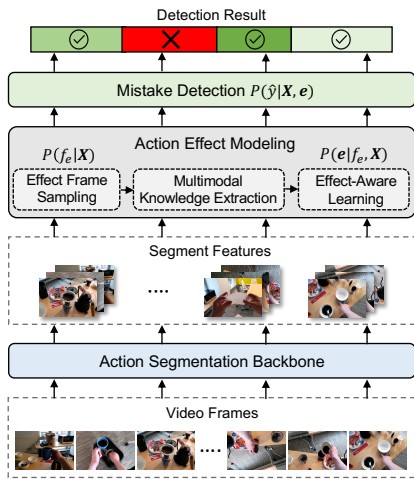

Figure 2: Framework overview.

## 3.3 ACTION SEGMENTATION

Following Lee et al. (2024) and Huang et al. (2025a), we adopt ActionFormer (Zhang et al., 2022) as the action segmentation backbone, which outputs a feature pyramid with multi-scale temporal resolutions. Inspired by prior work (Yang et al., 2024; Meng et al., 2020), we design a dynamic fusion module to refine temporal representations by adaptively aggregating the multi-scale features from the backbone, while preserving the hierarchical pyramid structure for segment boundary regression. This design aims to provide more discriminative segment features $\mathbf{X}$ for subsequent action effect modeling and mistake detection. Please refer to Appendix A for details about this module.

## 3.4 ACTION EFFECT MODELING

While the segment feature encodes the temporal dynamics of action execution, it is insufficient to capture the subtle yet critical cues manifested in the action's outcome. To bridge this gap, AEM integrates action effects by extracting multimodal features from selected effect frames and using them as external supervision. These features guide a learnable *effect token*, which distills outcome semantics into the action representation. As illustrated in Figure 3, the entire AEM pipeline consists of three steps: effect frame sampling, multimodal knowledge extraction, and effect-aware learning. It produces enriched effect-aware action representations that capture both execution and outcome.

**Effect Frame Sampling.** To identify the frame that best captures the outcome of a given action, we rank candidate frames within each action segment based on semantic relevance and visual clarity. For semantic relevance, motivated by Niu et al. (2024), we use GPT-4o (Hurst et al., 2024) to generate textual descriptions of the anticipated post-action states. Then we compute the frame-wise similarity between the segment's visual feature $\mathbf{X}$ and the description embeddings obtained from a pre-trained text encoder, followed by softmax normalization. For visual clarity, we apply the Laplacian operator to estimate the sharpness of each frame within the segment and normalize the scores between the $5th$ and $95th$ percentiles. The final ranking score combines both semantic relevance and visual clarity, and the top-ranked frame is selected as the effect frame.

**Multimodal Knowledge Extraction.** From the selected effect frame, we extract cues capturing both object states and spatial configurations, as they can reveal errors that occur in most procedural

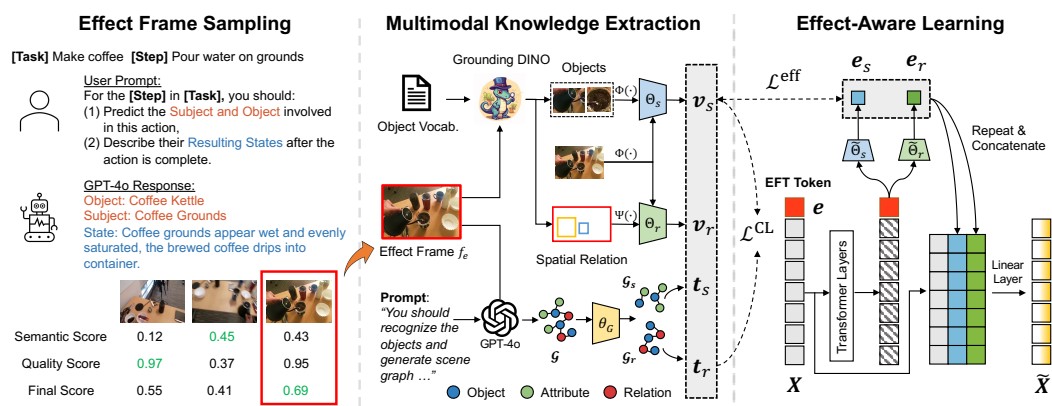

Figure 3: Overview of the Action Effect Modeling (AEM) module. Effect frame sampling and multimodal knowledge extraction are only used to learn effect state and relation projectors $\Theta_s, \Theta_r$ during training. Both of them are skipped during testing.

activities. We adopt a dual-branch strategy: a visual branch for grounded object-centric features, and a textual branch for symbolic scene abstraction.

In the visual branch, we use Grounding DINO (Liu et al., 2024) to detect relevant objects, then we extract object state and relation features $\mathbf{v}_s, \mathbf{v}_r$ by concatenating and projecting embeddings:

$$\mathbf{v}_s = \Theta_s \left( \Phi(f_e) \,\|\, (\Phi(f_e^{o_i})\big\|_{i=1}^{N}) \right), \quad \mathbf{v}_r = \Theta_r \left( \Phi(f_e) \,\|\, (\Psi(p_e^{o_i})\big\|_{i=1}^{N}) \right), \tag{2}$$

where $f_e$ is the effect frame, $f_e^{o_i}$ is the cropped $i$-th object regions, and $p_e^{o_i}$ is the corresponding coordinates of the object bounding boxes. $\Phi(\cdot)$ is a pre-trained image encoder, $\Psi(\cdot)$ is a positional encoding layer, and $\Theta_s, \Theta_r$ are two different multilayer perceptrons (MLPs). $\|$ denotes channel-wise concatenation, and $(\cdot)\big\|_{i=1}^{N}$ represents the successive concatenation of multiple embeddings.

Although visual grounding provides object-centric features that capture appearance and spatial cues from the effect frame, it lacks high-level semantic abstraction and relational reasoning. To address this, we introduce a textual scene graph branch to model the symbolic structure of the action effect. Given an effect frame and objects detected by Grounding DINO, we leverage the recognition and reasoning capabilities of a Multimodal Large Language Model (MLLM) by prompting GPT-4o to generate a scene graph $\mathcal{G} = (\mathcal{V}, \mathcal{E})$, where $\mathcal{V}$ contains three types of nodes: *object*, *relation*, and *attribute*, and $\mathcal{E}$ represents the set of directed edges. We encode the textual labels of all nodes in $\mathcal{V}$ using a pre-trained text encoder to obtain node embeddings, which are then fed into a Graph Neural Network (GNN) (Veličković et al., 2017) to compute context-aware features $\mathbf{t}_v$ for each node.

To abstract finer-grained effect information from text modality, we propose to decompose the generated scene graph $\mathcal{G}$ into two disentangled subgraphs: a state subgraph $\mathcal{G}_s = (\mathcal{V}_s, \mathcal{E}_s)$ and a relation subgraph $\mathcal{G}_r = (\mathcal{V}_r, \mathcal{E}_r)$. Specifically, the state subgraph includes the object nodes along with their adjacent attribute nodes, capturing object-level changes such as color or texture. The relation subgraph consists of the same object nodes and the intermediate relation node connecting them, representing spatial relationships such as "above" or "inside."Given $\mathbf{t}_v$ as the GNN feature of each node, we compute the textual features for object states and spatial relations by applying the following average pooling over subgraphs, where $|\mathcal{V}_s|$ and $|\mathcal{V}_r|$ are the number of nodes in each subgraph:

$$\mathbf{t}_s = \frac{\sum_{v \in \mathcal{V}_s} \mathbf{t}_v}{|\mathcal{V}_s|}, \ \mathbf{t}_r = \frac{\sum_{v \in \mathcal{V}_r} \mathbf{t}_v}{|\mathcal{V}_r|}, \tag{3}$$

**Effect-Aware Learning.** After acquiring multimodal effect features, a central challenge is how to effectively integrate these features into the action representation. A naive approach is to concatenate the effect and action segment features, but this would require querying MLLMs during inference, introducing prohibitive computational overhead. To circumvent this, we treat the effect features as supervision signals and employ a distillation-inspired strategy. Specifically, we introduce a learnable effect token that implicitly captures task-specific action effects through self-attention. During training, this token is aligned with the multimodal effect features to distill external knowledge into a

compact representation. Crucially, at inference time the framework relies only on the learned token, eliminating any dependency on external models to ensure efficient deployment.

As shown in Figure 3, we append the learnable effect token $\mathbf{e}$ to the segment-level features, which are then fed into Transformer encoding layers for temporal modeling. Subsequently, we align the projected effect tokens with the multimodal effect features in the state and relation latent spaces, respectively, which is formulated as:

$$\begin{aligned}
\mathcal{L}_s^{\text{eff}} &= ||\widetilde{\Theta}_s(\mathbf{e}) - \mathbf{v}_s||^2 + ||\widetilde{\Theta}_s(\mathbf{e}) - \mathbf{t}_s||^2, \\
\mathcal{L}_r^{\text{eff}} &= ||\widetilde{\Theta}_r(\mathbf{e}) - \mathbf{v}_r||^2 + ||\widetilde{\Theta}_r(\mathbf{e}) - \mathbf{t}_r||^2,
\end{aligned} \tag{4}$$

where $\widetilde{\Theta}_s(\cdot)$ and $\widetilde{\Theta}_r(\cdot)$ are spatial and relation projection layers, respectively. This enables efficient learning of a compact and generalizable effect-aware representation without relying on expensive effect-frame computation by MLLMs during inference time.

To enhance the consistency between the two supervision signals, we introduce a contrastive objective that explicitly aligns their visual and textual representations. While the effect loss encourages the learnable token to match each modality individually, this objective ensures that both modalities encode the same underlying semantics from complementary perspectives. Taking the object state as an example, we treat the visual and textual features $(\mathbf{v}_s, \mathbf{t}_s)$ as positive pairs, and all other samples in the batch as negatives. The contrastive loss is formulated as:

$$\mathcal{L}_s^{\text{CL}} = \sum_i \left[ -\log \frac{\exp\left(\cos(\mathbf{v}_s^i, \mathbf{t}_s^i)/\rho\right)}{\sum_j \exp\left(\cos(\mathbf{v}_s^i, \mathbf{t}_s^j)/\rho\right)} \right], \tag{5}$$

where $i$ and $j$ index samples in the batch, $\cos(\cdot)$ denotes the cosine similarity, and $\rho$ is a temperature coefficient. Similarly, we can calculate the relation contrastive loss as $\mathcal{L}_r^{\text{CL}}$. Together, these objectives enable the effect token to learn a compact multimodal representation of action effects, which is further integrated into the segment features as:

$$\widetilde{\mathbf{X}} = \mathcal{F}(\mathbf{X} \,\|\, \widetilde{\Theta}_s(\mathbf{e}) \,\|\, \widetilde{\Theta}_r(\mathbf{e})) \tag{6}$$

where $\|$ denotes channel-wise concatenation and $\mathcal{F}(\cdot)$ is a linear projection layer. The resulting fused segment features serve as a strong signal for downstream mistake detection.

## 3.5 MISTAKE DETECTION

Following previous work (Lee et al., 2024; Huang et al., 2025a), we adopt an one-class classification (OCC) setting for mistake detection, which learns canonical action patterns from correct samples only. Given the effect-enhanced segment features, we first perform an average-pooling over the enhanced effect-aware segment feature $\widetilde{\mathbf{X}}$ across temporal dimension to obtain the action embedding $\mathbf{x}_a$. Instead of learning a binary classifier, we introduce a prompt-based detector to increase task-specific discriminability. For each action label $a$, we construct a template-based textual prompt $\mathcal{P}_a$ (e.g., "An image showing [ACTION] for [TASK]"), and obtain its representation by:

$$\mathbf{y}_a = E\left(\mathbf{p} \,\|\, \text{Embed}(\text{Tokenize}(\mathcal{P}_a))\right), \tag{7}$$

where $\mathbf{p}$ is a learnable prefix embedding and $E(\cdot)$ is the frozen text encoder. $\|$ denotes concatenation.

During training, we align the action embedding with its corresponding prompt features while repulsing it from other prompt embeddings through a contrastive objective:

$$\mathcal{L}^{\text{det}} = -\frac{1}{B} \sum_{i=1}^{B} \log \left( \frac{\exp\left(\cos(\mathbf{x}_a^i, \mathbf{y}_a^i)/\rho\right)}{\sum_{c=1}^{C} \exp\left(\cos(\mathbf{x}_a^i, \mathbf{y}_a^c)/\rho\right)} \right), \tag{8}$$

where $\mathbf{x}_a^i$ denotes the $i$-th action embedding in a mini-batch $B$, and $C$ is the total number of actions in a specific task. During inference, we compute the similarity score $sim(\mathbf{x}_a, \mathbf{y}_a)$ between the action embedding and its corresponding textual prompt by $sim(\mathbf{x}_a, \mathbf{y}_a) = \cos(\mathbf{x}_a, \mathbf{y}_a)/\rho$. The final prediction of mistake probability can be obtained by $P(\hat{y} \mid \mathbf{X}, \mathbf{e}) = 1 - sim(\mathbf{x}_a, \mathbf{y}_a)$. The mistake detection result is determined by thresholding the mistake probability:

$$\hat{y} = \begin{cases} 1, & P(\hat{y} \mid \mathbf{X}, \mathbf{e}) > \tau, \\ 0, & \text{otherwise.} \end{cases} \tag{9}$$

where $\hat{y}$ is the predicted mistake label and $\tau$ is a predefined threshold.

Table 1: Mistake detection results on the EgoPER dataset (Lee et al., 2024). AUC and EDA are in percentage (%). Best results are **bold**, and second-best are underlined.

| Method | Quesadilla | | Oatmeal | | Pinwheel | | Coffee | | Tea | | All | |
| --- | --- | --- | --- | --- | --- | --- | --- | --- | --- | --- | --- | --- |
| | AUC | EDA | AUC | EDA | AUC | EDA | AUC | EDA | AUC | EDA | AUC | EDA |
| Random | 50.0 | 19.9 | 50.0 | 11.8 | 50.0 | 15.7 | 50.0 | 8.2 | 50.0 | 17.0 | 50.0 | 14.5 |
| HF$^2$-VAD | 62.6 | 34.5 | 62.3 | 25.4 | 52.7 | 29.1 | 59.6 | 10.0 | 62.1 | 36.6 | 59.9 | 27.1 |
| HF$^2$-VAD + SSPCAB | 60.9 | 30.4 | 61.9 | 25.3 | 51.7 | 33.9 | 60.1 | 10.0 | 63.2 | 35.4 | 59.6 | 27.0 |
| S3R | 51.8 | 52.6 | 61.6 | 47.8 | 52.4 | 50.5 | 51.0 | 16.3 | 57.9 | 47.8 | 54.9 | 43.0 |
| EgoPED | 65.6 | 62.7 | 65.1 | 51.4 | 55.0 | 59.6 | 58.3 | 55.3 | 66.0 | 56.0 | 62.0 | 57.0 |
| AMNAR | 71.9 | 61.4 | 75.4 | 65.0 | 65.4 | **65.0** | 67.8 | **73.5** | 61.9 | 57.0 | 68.5 | 64.4 |
| **Ours** | **80.8** | **68.1** | **77.0** | **68.6** | **69.9** | 61.2 | **70.3** | 66.4 | **71.1** | **69.4** | **73.8** | **66.7** |

Table 2: Mistake detection results on the CaptainCook4D dataset (Peddi et al., 2023).

| Method | Precision | AUC | EDA |
| --- | --- | --- | --- |
| Random | 44.9 | 51.2 | 49.7 |
| EgoPED | 56.5 | 54.9 | 69.8 |
| AMNAR | 66.8 | 60.2 | **72.3** |
| **Ours** | **68.1** | **62.5** | 71.9 |

## 3.6 TRAINING OBJECTIVE

Finally, we jointly optimize three objectives to train the model in an end-to-end fashion: the segmentation loss $\mathcal{L}^{\text{seg}}$ following Zhang et al. (2022), the action-effect modeling loss $\mathcal{L}^{\text{eff}}$ and $\mathcal{L}^{\text{CL}}$, and the mistake detection loss $\mathcal{L}^{\text{det}}$. The overall training objective is defined as:

$$\mathcal{L} = \mathcal{L}^{\text{seg}} + \mathcal{L}^{\text{eff}} + \mathcal{L}^{\text{CL}} + \mathcal{L}^{\text{det}}, \tag{10}$$

where $\mathcal{L}^{\text{eff}}$ and $\mathcal{L}^{\text{CL}}$ comprise both state and relation terms. By incorporating action-effect modeling into the training process, our framework learns to attend to semantically meaningful object state transitions and spatial relationships, thereby enhancing both representation and mistake detection.

## 4 EXPERIMENTS

### 4.1 EXPERIMENTAL SETUP

**Mistake Datasets.** We evaluate our method on two egocentric video datasets: EgoPER (Lee et al., 2024) and CaptainCook4D (Huang et al., 2025a). The EgoPER dataset consists of 385 egocentric cooking videos spanning 5 recipe categories, with a total duration of 28 hours. Among these, 213 videos contain only correct actions, while the remaining 178 include various execution mistakes. Following Lee et al. (2024), we use 80% of the normal videos for training, 10% for validation, and the remaining 10% together with all erroneous videos for testing. The CaptainCook4D dataset contains 384 videos covering 24 recipes, with a total length of 94.5 hours. Among all recordings, 164 are labeled as normal, while the remaining 220 videos include incorrect action steps. Following Huang et al. (2025a), we use all normal recordings for training and the erroneous ones for testing.

**Evaluation Metrics.** We follow prior work (Lee et al., 2024; Huang et al., 2025a) by using Area Under the Curve (AUC) and Error Detection Accuracy (EDA) to jointly measure model's ability to distinguish correct and erroneous actions within each video. AUC assesses frame-level discrimination between errors and non-errors, whereas EDA measures segment-level detection accuracy. Both metrics are computed by aggregating detection results across a densely sampled threshold range from 0 to 1, rather than relying on a single fixed threshold, enabling a more comprehensive evaluation of the model's behavior across the full spectrum of thresholds.

**Implementation Details.** We use off-the-shelf EVA-02 CLIP (Fang et al., 2024) for all image and text encoding. For the EgoPER dataset, we follow Lee et al. (2024) to use additional features based on Active Object Detection (AOD). For effect frame sampling, the final score is an average of the semantic relevance and visual quality scores. Following Lee et al. (2024) and Huang et al. (2025a), we

Table 3: Ablation studies (a–d), action segmentation results (e), and performance comparison with different scene graph generation models (f) on EgoPER dataset (Lee et al., 2024).

(a) Alignment between multimodal supervision with the projected state and relation effect tokens.

| AEM | $\mathcal{L}_s^{\text{eff}}$ Visual | $\mathcal{L}_s^{\text{eff}}$ Textual | $\mathcal{L}_r^{\text{eff}}$ Visual | $\mathcal{L}_r^{\text{eff}}$ Textual | AUC | EDA |
|---|---|---|---|---|---|---|
| ✗ | N/A | | | | 67.6 | 65.6 |
| ✓ | ✗ | ✗ | ✗ | ✗ | 67.9 | 65.8 |
| | ✓ | ✓ | ✗ | ✗ | 68.4 | 66.1 |
| | ✗ | ✗ | ✓ | ✓ | 69.4 | 66.3 |
| | ✓ | ✗ | ✓ | ✗ | 71.7 | 66.4 |
| | ✗ | ✓ | ✗ | ✓ | 69.9 | 66.0 |
| | ✓ | ✓ | ✓ | ✓ | **73.8** | **66.7** |

(b) Effect-frame sampling strategies.

| Method | AUC | EDA |
|---|---|---|
| w/o Effect | 67.6 | 65.6 |
| Last Frame | 70.6 | 65.7 |
| **Ours** | **73.8** | **66.7** |

(c) Dynamic cross-level interactions.

| Method | AUC | EDA |
|---|---|---|
| AMNAR | 68.5 | 64.4 |
| w/o Dyn | 71.8 | 63.4 |
| **w/ Dyn** | **73.8** | **66.7** |

(d) Alignment of supervisions in state and relation spaces.

| $\mathcal{L}_s^{\text{CL}}$ | $\mathcal{L}_r^{\text{CL}}$ | AUC | EDA |
|---|---|---|---|
| ✗ | ✗ | 66.8 | 64.7 |
| ✓ | ✗ | 69.9 | 64.5 |
| ✗ | ✓ | 72.6 | 65.1 |
| ✓ | ✓ | **73.8** | **66.7** |

(e) Action segmentation performance.

| Method | IoU | Edit | F1@0.5 | Acc |
|---|---|---|---|---|
| EgoPED | 44.6 | 61.3 | 47.5 | 68.5 |
| AMNAR | 56.3 | 69.4 | 57.3 | **75.3** |
| **Ours** | **58.5** | **69.7** | **58.5** | 73.5 |

(f) Performance comparison of our method with different scene graph generation models.

| Models | Quesadilla AUC | Quesadilla EDA | Oatmeal AUC | Oatmeal EDA | Pinwheel AUC | Pinwheel EDA | Coffee AUC | Coffee EDA | Tea AUC | Tea EDA | All AUC | All EDA |
|---|---|---|---|---|---|---|---|---|---|---|---|---|
| Qwen3-VL | 77.8 | **69.0** | **77.3** | **68.8** | **70.0** | **62.2** | 70.2 | 64.3 | **71.4** | 68.7 | 73.3 | 66.6 |
| GPT-4o | **80.8** | 68.1 | 77.0 | 68.6 | 69.9 | 61.2 | **70.3** | **66.4** | 71.1 | **69.4** | **73.8** | **66.7** |

independently train and test the model in each recipe and report the average of all recipes as the final performance. Both the effect-frame sampling and multimodal knowledge extraction are performed during the *data pre-processing* stage to avoid additional training overhead. All experiments were conducted with an NVIDIA RTX 6000 Ada GPU (48GB memory). More implementation details can be found in the Appendix A.

## 4.2 COMPARISON WITH STATE-OF-THE-ART

We compare our methods with video anomaly detection methods HF$^2$-VAD (Liu et al., 2021), HF$^2$-VAD + SSPCAB (Ristea et al., 2022) and S3R (Wu et al., 2022), as well as mistake detection methods EgoPED (Lee et al., 2024) and AMNAR (Huang et al., 2025a). As shown in Table 1, on the EgoPER dataset, our method outperforms in both metrics on most of the five tasks, with an average improvement of 5.3% on AUC and 2.3% on EDA. The results in Table 2 further demonstrate its effectiveness on the CaptainCook4D dataset, surpassing AMNAR by 2.7% in Precision and 2.3% in AUC. On both datasets, our EDA is occasionally lower than that of AMNAR. This gap is likely due to AMNAR's use of a dynamic programming block to construct task graphs that mitigate noise in action-segment labels. Since incorporating task graphs lies beyond the scope of our work, we leave it for future exploration.

## 4.3 ABLATION STUDIES

In the ablation study, we aim to answer the following questions:
1. Is Action Effect Modeling effective and generalizable?
2. Which is more important: object state effect or spatial relation effect?
3. How do visual and textual signals contribute to AEM?

4. Is aligning external multimodal features necessary?
5. How do different effect-frame sampling strategies perform?
6. Does multi-scale dynamic fusion module contribute to mistake detection?
7. Can open-source MLLMs replace the expensive closed-source models?

*1) Action Effect Modeling.* As shown in Table 3a, the first row reports the performance of baseline, which already achieves a comparable performance to AMNAR. This is attributed to the prompt-based detector's ability to capture task-specific action representations, thereby enhancing the discriminability of action execution. The $2^{nd}$ row further demonstrates that the learnable token can implicitly model action effects without relying on contextual knowledge provided by Grounding DINO or GPT-4o. After introducing different forms of external supervision, our model achieves consistent improvements, verifying its generalizability. Finally, when combining all multimodal effect supervision signals, our model achieves the best performance due to supervised effect modeling.

*2) State vs. Relation Effect.* The $3^{rd}$ and $4^{th}$ rows of Table 3a show the impact of state and relational effects. Compared to implicit token learning, explicitly introducing supervision signals for state or relational effects leads to performance gains, where relational supervision performs better, yielding a 1.0% improvement in AUC over state supervision. We argue that spatial relationships between objects provide more consistent and discriminative cues for action correctness. In contrast, object state changes are often subtle and visually ambiguous, making them more difficult to capture reliably.

*3) Visual vs. Textual Supervision.* The $5^{th}$ and $6^{th}$ rows of Table 3a report the impact of incorporating visual and textual features as external knowledge. Compared to baseline, using visual features improves AUC to 71.7% and EDA to 66.4%, while textual features yield a smaller gain. We argue that visual features are more effective because they directly capture object appearance and spatial layout, offering fine-grained cues to reflect execution-related visual differences. In contrast, textual features, although semantically informative, are abstracted from generated scene graphs and may introduce noise or miss subtle visual distinctions in complex egocentric environments.

*4) Effect Feature Alignment.* We further study the impact of aligning multimodal supervision features in state and relation effect representation spaces. As shown in Table 3d, using unaligned visual and textual supervisions leads to a suboptimal AUC of 66.8% and EDA of 64.7%, even lower than the model without effect modeling, highlighting the importance of cross-modal alignment. Introducing contrastive alignment in either the state space or the relation space improves AUC to 69.9% and 72.6%, improves EDA to 64.5% and 65.1%. The stronger gain from relation alignment suggests that object spatial relationships play a more critical role in modeling action effects. Finally, aligning both state and relation representations achieves the best AUC, demonstrating the complementary nature of these two effect types and the effectiveness of joint cross-modal supervision.

*5) Effect Frame Sampling.* To assess the effectiveness of our effect-frame sampling strategy, we compare it with a naive baseline that selects the last frame of each action segment, assuming that action effects are most visible at the end. As shown in Table 3b, this heuristic raises AUC to 70.6%, confirming that incorporating effect information benefits mistake detection. In contrast, our proposed strategy, which jointly considers semantic relevance and visual clarity further, boosts AUC to 73.8% and EDA to 66.7%. These results imply that a more informed selection of effect frames yields more discriminative representations, thereby improving detection performance.

*6) Dynamic Fusion Module.* Table 3c evaluates whether enhancing multi-scale temporal representations contributes to the final detection performance. The results show that incorporating dynamic fusion yields a 2% improvement in AUC and a 3.3% gain in EDA, highlighting the importance of expressive action segment representations for downstream mistake detection. Notably, even without this module, our model still surpasses AMNAR by 3.3% AUC and achieves comparable EDA, further highlighting the superiority of our overall framework.

*7) Alternative open-source MLLM.* In this work, we employ a state-of-the-art closed-source MLLM GPT-4o (Hurst et al., 2024) to generate action scene graphs that provide supervision signals for our method. To assess whether alternative models can fulfill the same role, we replace GPT-4o with the latest open-source MLLM Qwen3-VL (30B) (Bai et al., 2025). As shown in Table 3f, the scene graphs generated by Qwen3-VL yield mistake detection performance comparable to that by GPT-4o, indicating that open-source models can serve as a cost-effective alternative for the generation of scene graphs while maintaining strong performance.

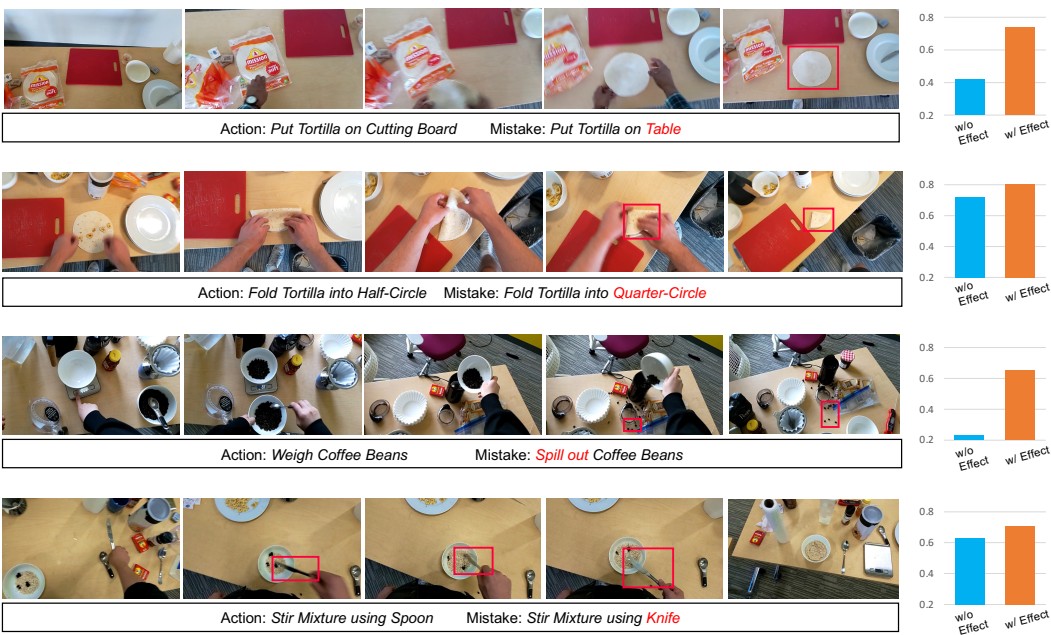

Figure 4: Examples of mistakes occurring in different actions. The right bar charts show mistake probabilities predicted by models without (in blue) and with (in orange) effect modeling. Red boxes in images are only used to highlight mistake regions for clearer visualization.

## 4.4 ACTION SEGMENTATION

Table 3e presents a comparison of action segmentation performance. All models, including ours, use ActionFormer (Zhang et al., 2022) as the segmentation backbone, ensuring a fair comparison. Our approach outperforms prior methods across most evaluation metrics, highlighting its effectiveness in enhancing action segmentation in addition to mistake detection. These results also indicate that our framework holds promise for broader video understanding tasks where action effects serve as critical cues, such as action recognition (Bao et al., 2021) and action analysis (Huang et al., 2025b).

## 4.5 QUALITATIVE ANALYSIS

Figure 4 shows visualizations of the action segments and the prediction of mistake probabilities. The results highlight two key advantages of our approach: (i) the model effectively detects errors that appear in the final outcomes through action–effect modeling (row 1 to row 3), and (ii) it can also identify execution errors that arise during the process enabled by the prompt-based detector, even when the visual outcome seems correct (row 4). Together, these findings demonstrate the complementarity between effect-aware representations and temporal execution modeling in capturing a broader spectrum of procedural errors. More visualization can be found in the Appendix C.

## 5 CONCLUSION

We presented a unified framework for procedural mistake detection that explicitly models the relationship between action execution and its resulting effect. The proposed Action Effect Modeling (AEM) enriches action representations with effect-aware cues by integrating object states and spatial relationships, guided through multimodal supervision. Combined with a prompt-based detector, our framework effectively captures both subtle execution errors and outcome discrepancies, achieving substantial improvements over prior methods on mistake detection benchmarks. Future work includes extending the framework to spatial-temporal action effect modeling for long-range procedural reasoning and improving mistake interpretability via human-understandable explanations generated by Large Language Models (LLMs).

ACKNOWLEDGMENT

This research was partially sponsored by the MSU Jenison Fund and Army Research Office (under Grant Number W911NF-24-1-0385). The views and conclusions contained in this document are those of the authors and should not be interpreted as representing the official policies, either expressed or implied, of the Army Research Office or the U.S. Government. The U.S. Government is authorized to reproduce and distribute reprints for Government purposes notwithstanding any copyright notation herein. The authors thank Wentao Bao for insightful suggestions on the methodological aspects of this work.

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

APPENDIX

This appendix is organized as follows:

- **Appendix A** introduces more details about the implementation.
- **Appendix B** presents more experimental results and analysis.
- **Appendix C** provides more visualizations for demonstration.
- **Appendix D** shares further discussion on limitations and future directions.
- **Appendix E** clarifies the use of the large language model.

## A    IMPLEMENTATION DETAILS

### A.1    DYNAMIC FUSION MODULE

As described in Section 3, we design a simple yet effective dynamic fusion module to bridge inter-actions among multi-scale frame features produced by the ActionFormer backbone, thereby refining temporal representations. The internal structure of this module is illustrated in Figure 5. Frame features at each level, with different temporal lengths, are first passed through a convolutional layer followed by a normalization layer to capture temporal information at varying granularities. For each feature, the features from its adjacent levels are up-sampled or down-sampled along the temporal dimension via linear interpolation to match its length, after which a weighted sum is applied. The weights are learnable and specific to each feature. The resulting features preserve their original length and dimensionality while being enriched with aggregated temporal information. An ablation study in Table 3e validates the effectiveness of this dynamic fusion module for mistake detection.

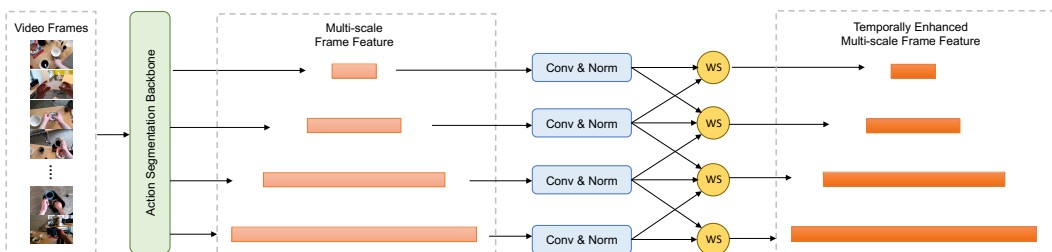

Figure 5: Structure of the dynamic fusion module. Each *Conv&Norm* block represents a convolu-tional layer followed by a normalization layer, and each *WS* block denotes a weighted-sum operation.

### A.2    EFFECT-FRAME SAMPLING

In this work, we perform action-effect modeling by leveraging the effect knowledge embedded in effect frames. To sample effect frames within action segments, we first predict key objects that indicate the action effect and then apply semantic scoring to evaluate the relevance of each video frame to the effect. Figure 7a presents the complete GPT-4o prompt used for key-object prediction and generation of textual effect descriptions.

### A.3    ACTION SCENE GRAPH

In this work, we construct a symbolic action scene graph from sampled effect frames to generate textual supervision for effect learning. This subsection details the implementation of the action scene graph, including: (1) generating the scene graph by prompting GPT-4o and parsing its output, (2) decomposing the graph into object-state and spatial-relation subgraphs, and (3) presenting a concrete example that illustrates the overall process.

### A.3.1    SCENE GRAPH CONSTRUCTION

We prompt the powerful GPT-4o to recognize the action scene and output the scene information. Figure 7b shows the complete prompt we used. Scene information is returned in the format of the

dictionary (JSON file), and we design algorithms to parse the output and construct the action scene graph based on it. Algorithm 1 describes the graph construction process, where the input includes the object relation dictionary (*Relation_Dict*) and the object attribute dictionary (*Attribute_Dict*) generated by the VLM. The output consists of two dictionaries: one containing the graph nodes categorized by type (*Nodes*) and the other storing the edges connecting pairs of nodes (*Nodes*).

### A.3.2    ACTION SCENE GRAPH DECOMPOSITION

Algorithm 2 describes the graph decomposition process, where the input includes *Nodes* and *Nodes* obtained from graph construction, as well as the specified action-related objects generated by effect frame sampling. The outputs are separate sub-graphs of object state (*S-Graph*) and object spatial relationships (*R-Graph*).

### A.3.3    EXAMPLE OF SCENE GRAPH PROCESSING

Figure 6 illustrates how an action scene graph is constructed and decomposed for the effect frame of *Pouring Water on Coffee Ground* within the task of *Making Coffee*.

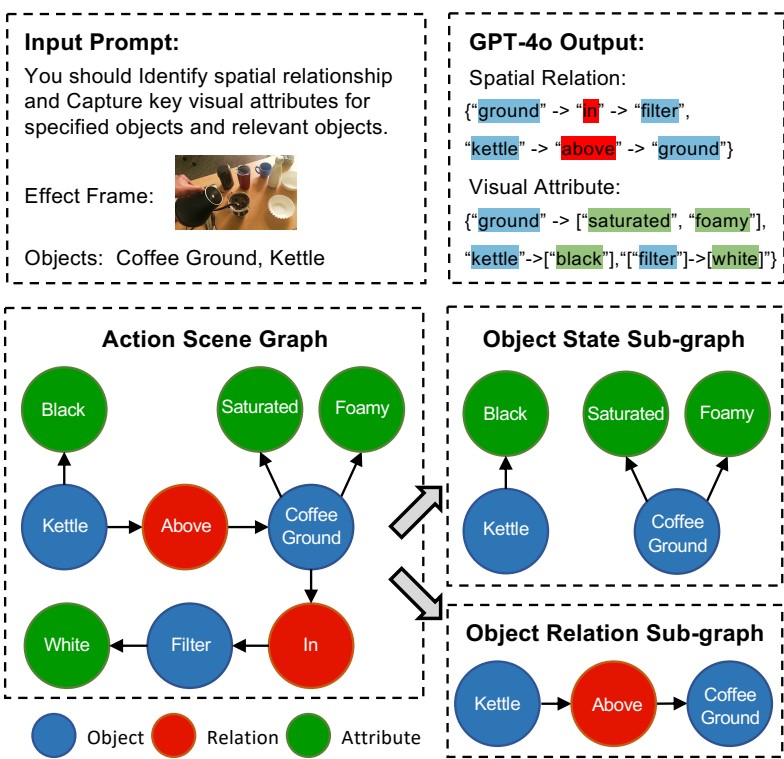

Figure 6: An example showing the construction and decomposition of action scene graph.

### A.4    SOLUTION TO FAILED KNOWLEDGE EXTRACTION

During effect-frame preprocessing for multimodal feature extraction, several failure cases were observed. Grounding DINO may miss relevant objects or yield low-confidence detections, while GPT-4o may fail to recognize the image, leading to missing object states or spatial relations. To address these issues, we introduce an Effect Mask. The mask is assigned a value of $1$ only when both Grounding DINO and GPT-4o successfully process the effect frame; otherwise, it is set to $0$. During feature alignment, only instances with a mask value of $1$ contribute to loss computation and are used to generate enhanced segment features for mistake detection. For masked-out instances, effect modeling is bypassed, and segment features from the segmentation backbone are directly fed into the mistake detector.

---

**Algorithm 1** Action Scene Graph Construction

---

**Input:** *Relation_Dict*, *Attribute_Dict*
**Output:** *Nodes*, *Edges*
    *// Initialization.*
 1: *Nodes* ← dict("Object":[], "Attribute":[], "Relation":[])
 2: *Edges* ← list([])

    *// Add object-relation edges.*
 3: **for** each $(obj_A, rel, obj_B)$ in *Relation_Dict* **do**
 4:     **if** $obj_A \notin Nodes["Object"]$ **then**
 5:         Append $obj_A$ to *Nodes*["Object"]
 6:     **end if**
 7:     **if** $obj_B \notin Nodes["Object"]$ **then**
 8:         Append $obj_B$ to *Nodes*["Object"]
 9:     **end if**
10:     **if** $rel \notin Nodes["Relation"]$ **then**
11:         Append $rel$ to *Nodes*["Relation"]
12:     **end if**
13:     Append $(obj_A, rel)$, $(rel, obj_B)$ to *Edges*
14: **end for**

    *// Add object-attribute edges*
15: **for** each $(obj, attr)$ in *Attribute_Dict* **do**
16:     **if** $obj \notin Nodes["Object"]$ **then**
17:         Append $obj$ to *Nodes*["Object"]
18:     **end if**
19:     **if** $attr \notin Nodes["Attribute"]$ **then**
20:         Append $attr$ to *Nodes*["Attribute"]
21:     **end if**
22:     Append $(obj, attr)$ to *Edges*
23: **end for**

24: **return** *Nodes*, *Edges*

---

**Algorithm 2** Action Scene Graph Decomposition

---

**Input:** Extracted graph $Nodes$, graph $Edges$, and key objects $Objs = \{obj_i\}_{i=1}^{N}$
**Output:** State graph $S\text{-}Graph$, relation graph $R\text{-}Graph$
    *// Initialization.*
 1: Initialize $S\text{-}Graph \leftarrow$ dict({"Nodes":[], "Edges":[]})
 2: Initialize $R\text{-}Graph \leftarrow$ dict({"Nodes":[], "Edges":[]})

    *// Construct the state graph from object-attribute edges*
 3: **for** each edge $(a, b)$ in $Edges$ **do**
 4:     **if** $a \in Objs$ **and** $b \in Nodes["Attribute"]$ **then**
 5:         Append $a$ and $b$ to $S\text{-}Graph["Nodes"]$
 6:         Append $(a, b)$ to $S\text{-}Graph["Edges"]$
 7:     **end if**
 8: **end for**

    *// Construct the state graph from object-attribute edges*
 9: **for** each edge $(a, b)$ in $Edges$ **do**
10:     **if** $a \in Objs$ **and** $b \in Nodes["Relation"]$ **then**
11:         Append $a$ and $b$ to $R\text{-}Graph["Nodes"]$
12:         Append $(a, b)$ to $R\text{-}Graph["Edges"]$
13:     **end if**
14: **end for**

15: **return** $S\text{-}Graph, R\text{-}Graph$

---

For the following step in process of [goal], you should:
(1)  predict relevant objects,
(2)  describe resulting states of relevant objects after the action is complete. Use three concise
     sentences. Do not use the verb in [step].

Example A:
[goal]: Make Kimchi Fried Rice [step]: add ham
Objects: [ham, fried rice, pan]
Descriptions:
-   The diced ham is mixed with fried rice.
-   The ham is on the pan.
-   The pan contains ham.

Example B:
[goal]: Make Pancakes [step]: pour egg
Objects: [egg, batter, bowl]
Descriptions:
-   The egg is mixed with the pancake batter.
-   The egg is in the mixing bowl.
-   The pancake batter contains egg.

[goal]: {task} [step]: {step}

(a) Prompt used to generate relevant objects and resulting outcome descriptions.

You are an expert in spatial reasoning and visual scene understanding. Given an image, your task is to:
1. Recognize all objects in the image.
2. Identify the spatial relationship between objects in the image.
3. Capture key attributes of objects.
4. Generate a structured scene graph that accurately represents the real-world state in the image.

Important Guidelines:
1. Describe spatial relations using clear spatial terms: ("above", "below", "on", "under",   "to the left
of", "to the right of", "next to", "in front of", "behind").
2. Ensure each object's attributes (e.g., shape, color, material) are captured accurately.
3. Only describe what you see in the image; do not infer beyond the visual evidence.

Input Format (Image and Text):
Image: (Provided image)
Text: Generate a scene graph for the objects in the image, focusing on their relationships and attributes.

Output Format (JSON - Relations + Attributes + Sentences):
```
{
   "objects": ["<object_1>", "<object_2>", "<object_3>", ...],
   "relation": [
      {"subject": "<object_1>", "relation": "<relation>", "object": "<object_2>"},
      {"subject": "<object_1>", "relation": "<relation>", "object": "<object_3>"},
      {"subject": "<object_2>", "relation": "<relation>", "object": "<object_3>"},
      ...
   ],
   "attribute": [
      {"subject": "<object_1>", "attribute": ["<attribute_1>", "<attribute_2>"]},
      {"subject": "<object_2>", "attribute": ["<attribute_1>", "<attribute_2>"]},
      ...
   ]
}
```

(b) Prompt used to recognize object attributes and spatial relationships.

Figure 7: Prompts used in this work.

# B EXPERIMENTS

## B.1 IMPACT OF VLM ON PERFORMANCE

Table 4: Performance by using different VLMs for action scene generation on EgoPER dataset.

| Model | Quesadilla AUC | EDA | Oatmeal AUC | EDA | Pinwheel AUC | EDA | Coffee AUC | EDA | Tea AUC | EDA | All AUC | EDA |
|---|---|---|---|---|---|---|---|---|---|---|---|---|
| Qwen3-VL | 77.8 | 69.0 | 77.3 | 68.8 | 70.0 | **62.2** | 70.2 | 64.3 | **71.4** | 68.7 | 73.3 | 66.6 |
| GPT-4o | **80.8** | 68.1 | 77.0 | 68.6 | 69.9 | 61.2 | 70.3 | 66.4 | 71.1 | 69.4 | 73.8 | 66.7 |
| GPT-5 | 78.9 | **69.1** | **79.1** | **70.1** | **70.4** | 61.9 | **70.4** | **66.4** | 71.3 | **69.8** | **74.0** | **67.5** |

At the time of completing the main paper, we employed the SOTA closed-source VLM GPT-4o (Hurst et al., 2024) and the open-source VLM Qwen3-VL (Bai et al., 2025) to generate action scene graphs. In this subsection of the Appendix, we extend Table 3f in the main paper by providing further evaluation results of our proposed method using a more recent GPT-5 model (OpenAI, 2025). Table 4 shows that the GPT-5 variant achieves the best detection performance which exceeds both models, suggesting that our method continues to benefit from ongoing advances in VLMs.

## B.2 ACTION SEGMENTATION

Table 5: Impact of predicted vs. ground-truth action segmentation on EgoPER dataset.

| Action Seg | Quesadilla AUC | EDA | Oatmeal AUC | EDA | Pinwheel AUC | EDA | Coffee AUC | EDA | Tea AUC | EDA | Average AUC | EDA |
|---|---|---|---|---|---|---|---|---|---|---|---|---|
| Pred | 80.8 | 68.1 | 77.0 | 68.6 | 69.9 | 61.2 | 70.3 | 66.4 | 71.1 | 69.4 | 73.8 | 66.7 |
| GT | 92.8 | 78.4 | 94.2 | 75.0 | 80.7 | 72.0 | 80.8 | 71.3 | 84.8 | 76.7 | 86.7 | 74.7 |

To assess the impact of action segmentation on mistake detection, we evaluated our method by replacing predicted segments with ground-truth segments *during inference* on the test set. As shown in Table 5, using ground-truth segmentation yields a significant improvement in mistake detection performance, which can be regarded as an upper bound for the current framework. This result suggests that improvements in segmentation can consistently enhance detection performance; therefore, future work will explore leveraging more advanced action segmentation models.

## B.3 PROMPT TUNING IN MISTAKE DETECTOR

Table 6: Ablation results for the learnable prompts on EgoPER dataset.

| Learnable | Quesadilla AUC | EDA | Oatmeal AUC | EDA | Pinwheel AUC | EDA | Coffee AUC | EDA | Tea AUC | EDA | All AUC | EDA |
|---|---|---|---|---|---|---|---|---|---|---|---|---|
| ✗ | 77.4 | **68.9** | 76.6 | 57.5 | 69.5 | 60.7 | 68.5 | **67.0** | **72.1** | 63.6 | 72.8 | 63.5 |
| ✓ | **80.8** | 68.1 | **77.0** | **68.6** | **69.9** | **61.2** | **70.3** | 66.4 | 71.1 | **69.4** | **73.8** | **66.7** |

In the mistake detector, we adopt learnable prompts combined with prefix-tuning techniques (Li & Liang, 2021) to capture action patterns. To evaluate the effectiveness of the prompt-based detector, we compare learnable prompts with fixed prompts. Both methods use the same template, "an image showing [ACTION] for [TASK]", to generate contextual embeddings. The fixed prompt does not include learnable embeddings; instead, it directly uses the frozen text encoder of EVA-02 CLIP (Fang et al., 2024) to extract features. As shown in Table 6, the learnable prompt achieves superior performance on most subtasks as well as on average compared to the fixed prompt. This indicates that fixed textual descriptions, due to the lack of temporal and contextual motion encoding, fail to align effectively with visual action features. Therefore, additional learnable embeddings are necessary to capture the temporal dynamics of actions.

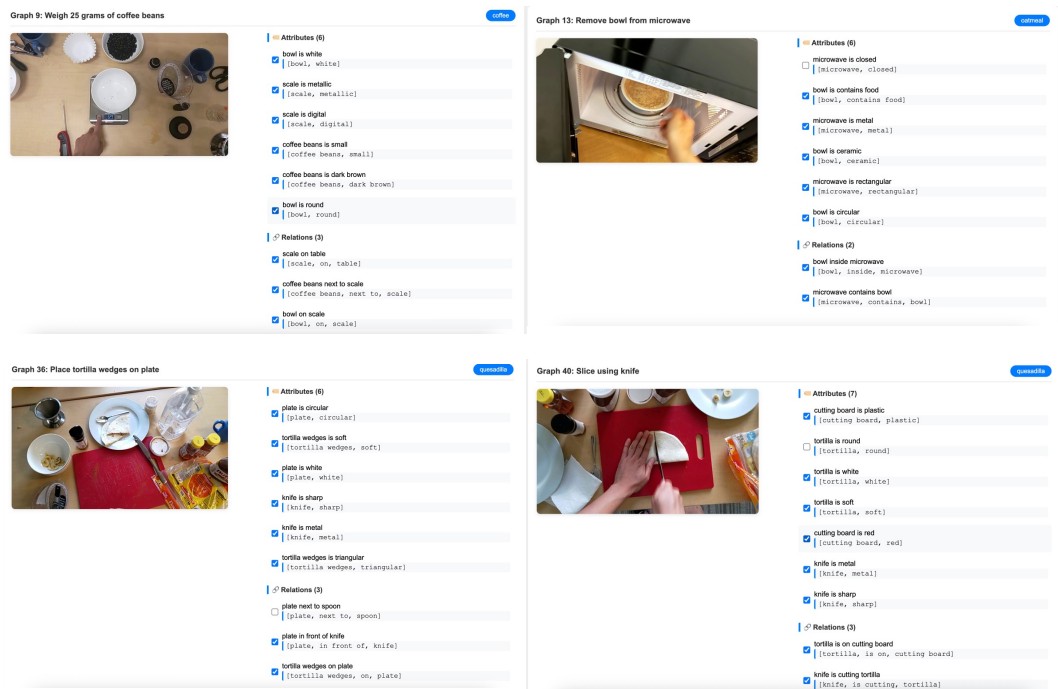

Figure 8: Web interface for scene recognition manual check.

For tasks where the performance of learnable prompts is slightly lower than that of fixed prompts, such as *Make Quesadilla*, we observed that many erroneous actions and their corresponding mistakes are less related to temporal dynamics. For example, *Place tortilla on table* (instead of on cutting board) and *Fold tortilla into a quarter-circle* (instead of a half-circle) already encode sufficient object-related information by using their textual labels. In such cases, our effect-aware model can distinguish the errors effectively, and introducing additional learnable prompts to capture action dynamics brings limited benefit and may even interfere with effect encoding, leading to performance degradation. As this work provides an initial exploration of encoding action temporal dynamics in mistake detection, extending it to a broader range of action labels is an interesting research direction.

### B.4 Effect Frame Scene Recognition

In this work, we employ GPT-4o (Hurst et al., 2024) to recognize the environments of effect frames, which are then used to build scene graphs and extract symbolic knowledge. Recognition errors in this process may introduce noise into the effect modeling. Although the main paper demonstrates that effect modeling substantially improves mistake detection, we further conducted a quantitative evaluation of GPT-4o's recognition performance on complex scenes.

Specifically, we randomly sampled 20 recognition results from each sub-dataset of the EgoPER dataset (Lee et al., 2024), yielding 100 scene instances in total. As shown in Figure 8, we developed an HTML-based interface to display the recognized attributes and spatial relations for each effect frame. Two graduate students from the computer science department, with no domain conflicts, were recruited and randomly assigned 50 scenes each for manual evaluation. We define two metrics to assess recognition performance: Success Rate, the proportion of effect frames in which all recognition results are correct, and Accuracy, the proportion of correctly recognized items among all recognition results. The final scores were obtained by averaging the results of the two annotators.

Manual evaluation yielded a success rate of 48.4% and an accuracy of 87.0%. Although GPT-4o occasionally produces recognition errors, its multi-perspective scene descriptions generally complement one another, enabling reliable reconstruction of object states and spatial relationships. This provides a solid foundation for effect modeling, and we anticipate that such limitations will diminish as multimodal large language models continue to advance.

## C    VISUALIZATIONS

### C.1    ACTION SCENE GRAPH

Figure 9 demonstrates examples of action scene graph build upon GPT-4o recognition and parsed by Algorithm 1. They captures attributes and spatial relationship of objects in a structured manner.

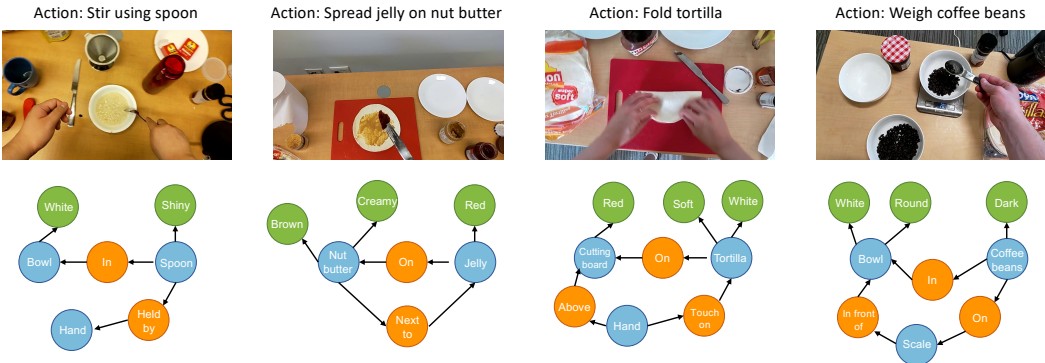

Figure 9: Examples of action scene graphs built upon effect frames.

### C.2    ACTION SEGMENT FRAME SCORING

Figure 10 shows examples of the top-3 video frames with the highest weighted scores in effect-frame sampling. In our implementation, semantic and quality scores are averaged to produce the final weights. Visual inspection indicates that these frames, which are often consecutive in the video, contain highly similar content. Empirically, we find that using the top-$K$ frames instead of only the top-1 provides negligible performance gains while substantially increasing annotation and training costs. When $K$ is larger (e.g., $K = 5$), lower-scoring frames may introduce noisy or mismatched content, leading to reduced mistake-detection performance. Therefore, we select only the highest-scoring frame as the effect frame in our sampling strategy.

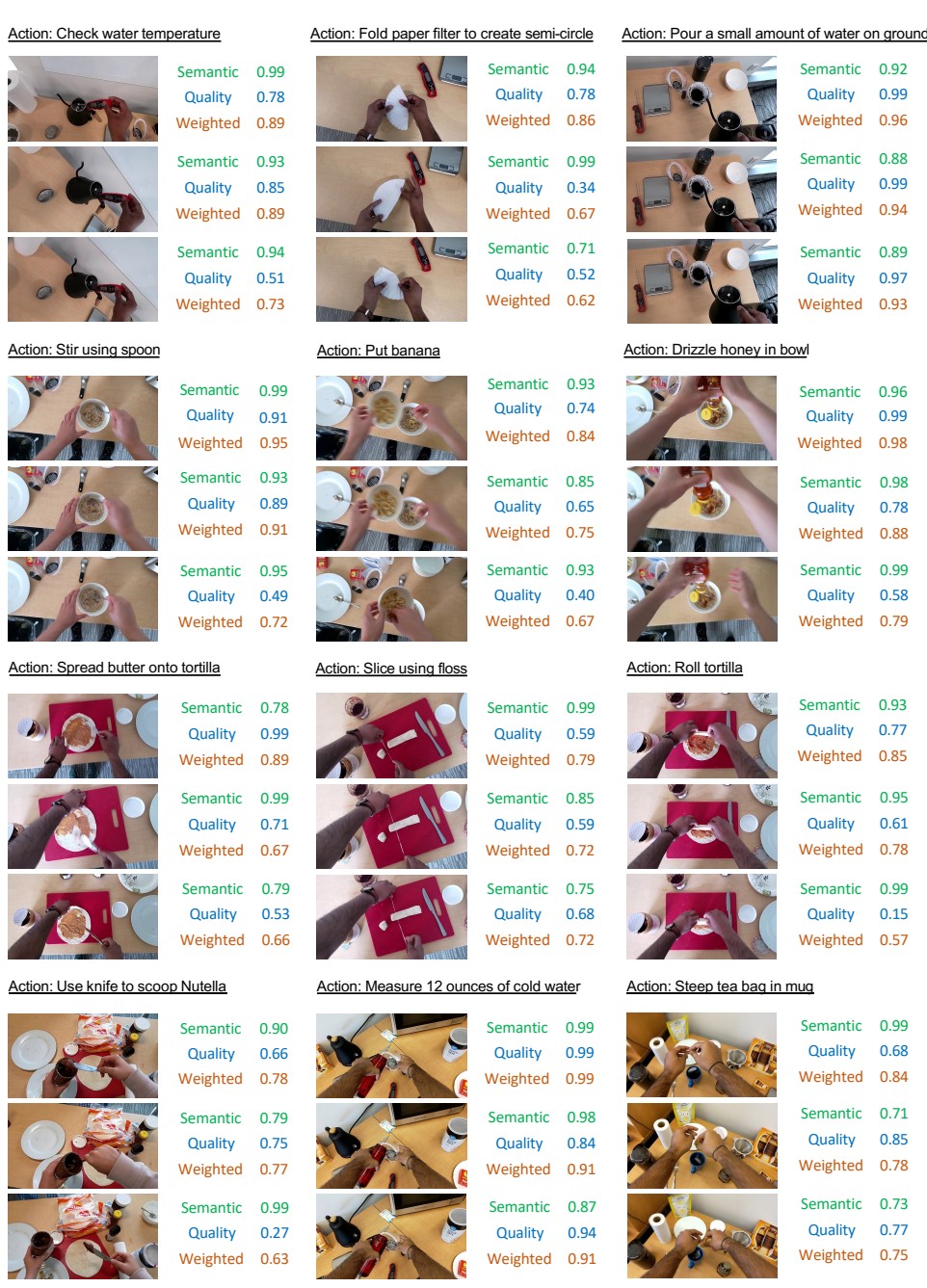

Figure 10: Examples of scores by effect frame sampling for different actions.

## D  FURTHER DISCUSSION

**Dependence on object detection and scene graphs.**    The performance of our framework remains tied to the accuracy of object detection and scene graph generation. In egocentric and cluttered environments, objects are frequently occluded, blurred, or partially visible, leading to unstable detection and relationship modeling, which in turn degrades the reliability of error detection. Future directions include incorporating stronger spatiotemporal representations, such as 3D or 4D modeling, together with multimodal cues, such as depth maps or point clouds, to improve robustness under occlusion and enhance object localization and relational reasoning.

**Spatial-temporal modeling.**    Current effect modeling focuses on single-step actions and does not explicitly capture long-range dependencies across multiple steps. In real procedural tasks, small early deviations can accumulate and lead to significant downstream errors. Future directions include modeling cross-step dependencies through hierarchical task graphs, causal reasoning structures, or program-level state transition models, thereby extending the framework to long-horizon, multi-step reasoning and enabling more comprehensive detection of complex procedural errors.

**Supervision of effect modeling.**    Our framework currently relies on multimodal supervision generated by large pretrained models, which may hinder generalization to new domains or resource-constrained settings. To reduce this dependency, future work may explore weakly supervised or self-supervised signals, for example, leveraging temporal consistency, action outcome comparisons, or environment state transitions to automatically construct supervision, thereby improving scalability in broader domains and scenarios.

**Dependence on large models.**    Our method relies on large models to generate pseudo-labels or auxiliary signals for action effect modeling, yet their outputs can be stochastic and inconsistent. For instance, in experiments, we observed that an identical effect frame may yield slightly different spatial relationship descriptions across runs, and such noise can propagate into learned effect representations and impact training stability. Potential mitigation strategies include adopting open-source models with reproducible checkpoints and fixed random seeds, as well as introducing consistency constraints, cross-model voting, or automatic filtering mechanisms to reduce noise propagation.

**Interpretability of mistakes.**    Most existing methods, including ours, formulate error detection as binary classification, which provides limited insight into the underlying causes of errors. Enhancing interpretability therefore represents an important research direction. One possible solution is to integrate LLMs or LVLMs to generate human-understandable explanations, improving the transparency and usability of error detection systems.

**Potential of world models.**    World models have recently emerged as an active research direction, as they capture environment dynamics and predict future scenes based on historical observations. These capabilities make them promising for procedural video mistake detection, particularly for online and one-class classification (OCC) settings. Future directions include predicting future states in pixel or latent space conditioned on past observations and comparing them with actual outcomes to identify inconsistencies, which correspond to procedural errors.

## E  USE OF LARGE LANGUAGE MODEL

In this work, large language models (LLMs) were employed solely for English translation and language polishing, with the goal of improving fluency and clarity of the manuscript so that the contributions can be presented to readers more effectively. LLMs were **not** used for idea generation, conceptualization, or any other aspect of the research.

