# OpenReview forum: "Procedural Mistake Detection via Action Effect Modeling"
_ICLR.cc/2026/Conference — ICLR 2026 Poster_

### Official Review · Reviewer_JzSx · 2025-10-21

**Soundness:** 2
**Presentation:** 3
**Contribution:** 2
**Rating:** 2
**Confidence:** 3

**Summary:**

Procedural Mistake Detection is important for supporting learning and task execution. This paper proposed Action Effect Modeling (AEM) to capture not only the action execution but also its outcome.
The AEM module consists of:
1) Effect frame sampling: Select the most informative effect frame based on semantic score and quality score.
2) Multimodal knowledge extraction: By leveraging the foundation model, Grounding DINO, for object detection to get the object states and relations. Then, use GPT-4o to generate the scene graph and decompose into a state subgraph and a relation subgraph. This process generate extracts the complementary cues.
3) Effect-aware learning: To avoid querying multimodal large language models during inference, a learnable effect token (EFT) captures task-specific action effects through self-attention so that during inference time, the framework depends only on this learned token.

AEM achieves state of the art performance in 2 benchmarks, showcasing the importance of modeling both execution and outcome in mistake detection.

**Strengths:**

The main strength in this paper is to provide a in-depth ablation studies to show that 1) AEM is effective and generalizable, 2) spatial relationships provide more consistent and discriminative cues for action correctness, 3) Visual features are more effective, 4) cross-modal alignment is important, 5) Simple last frame selection is not good enough, need to incorporate semantic relevance and visual clarity, 6) expressive action segment representations are important downstream mistake detection.

This ablation study is very thorough, and unlike the ablation studies in most paper, this paper yields new interesting "general" insights that may be transferable to other domains beyond action effect modeling. Additionally, the implementation details are clear in the paper including models and compute budgets.

The figures are well-made for understanding the method.

**Weaknesses:**

1. The mistake detection result is determined by a predefined threshold. This poses a major problem, if you change the predefined threshold, the results will be greatly affected? On the other hand, if you try different values of the threshold to maximize the performance, it is essentially "tuning" on the test set.
2. The modular approach, despite effectiveness, it may introduce problems like longer processing times, extra engineering, additional training.
3. No baselines of strong end-to-end reasoning models like OpenAI o3, o4 and GPT-5.
4. The idea of modeling both action and effect is not very novel as it has been studied in many earlier papers with different focuses (multimodal action effect prediction, state change in videos, action-conditioned world models) [1,2,3].

References:
[1] Dagan et al. Learning the Effects of Physical Actions in a Multi-modal Environment. 2023.
[2] Souček et al. Look for the Change: Learning Object States and State-Modifying Actions from Untrimmed Web Videos. 2022
[3] Hu et al. Video Prediction Policy: A Generalist Robot Policy with Predictive Visual Representations. 2024

**Questions:**

1. What is the predefined threshold used, as mentioned in the Weakness 1?

---

> ### Author Response · Authors · 2025-11-26
> **Response to Reviewer JzSx**
>
> Thank you for your constructive feedback. We provide responses addressing each of your concerns in detail.
>
> **W1 & Q1: The mistake detection result is determined by a predefined threshold.**
>
> We clarify that we do not tune the threshold on the test set. Instead, following the evaluation protocol used in previous work [1,2], we compute EDA and AUC by aggregating detection results over a densely sampled range of thresholds from 0 to 1 with a step size of 0.025. This protocol does not adjust the threshold based on test-set performance, but comprehensively evaluates the model’s behavior across a spectrum of thresholds.
>
> **W2: The modular approach may introduce problems like longer processing times, extra engineering, additional training.**
>
> | Model | Training speed (sec/step) ↓ | Inference speed (sec/video) ↓ |
> |:----:|:---------------------------:|:------------------------------:|
> | AMNAR | 0.94 | 0.29 |
> | Ours  | 0.28 | 0.12 |
>
> *Table: Training and inference time comparison.*
>
> Although our framework is modular by design, all components are trained jointly under a single end-to-end objective, so no additional stage-wise training or extra engineering pipelines are required. To address the concern about processing time, we report both the average time per training step, and the average inference time per video, comparing our method with a prior approach [2]. The results in the table above show that our model remains efficient in practice.
>
> **W3: Strong end-to-end reasoning models as baselines.**
>
> | Model | Quesdilla | Oatmeal | Pinwheel | Coffee | Tea | Average |
> |:-----:|:---------:|:-------:|:---------:|:------:|:---:|:-------:|
> | GPT-5 | 24.7 | 26.7 | 25.7 | 37.2 | 21.1 | 27.1 |
> | Gemini 2.5 Pro | 28.3 | 27.0 | 27.5 | 41.1 | 28.1 | 30.4 |
> | **Ours** | **68.1** | **68.6** | **61.2** | **66.4** | **69.4** | **66.7** |
>
> *Table: Model performance on EgoPER Dataset.*
>
> We employ GPT-5 [3] and Gemini 2.5 Pro [4] to perform mistake detection and evaluate their performance on the EgoPER dataset [1]. We report and compare their Error Detection Accuracy (EDA). As shown in the table above, our method achieves higher performance than both large models, indicating that the proposed action effect modeling yields stronger mistake recognition capability than generic end-to-end reasoning.
>
> **W4: The idea of modeling both action and effect has been studied in many earlier paper.**
>
> We thank the reviewer for pointing out the related literature [5,6,7], which indeed study action–effect relations but with different goals and usages. Dagan et al. [5] predict symbolic next object states from state–action pairs and evaluate effect prediction accuracy. Souček et al. [6] discover object state changes and their temporal localization in web videos, assuming actions are correctly executed. Hu et al. [7] encode effects implicitly as future visual predictions within a diffusion backbone and use them for control, rather than explicit reasoning about success or failure.
>
> In contrast, our work builds an explicit task-aware and object-centric action–effect representation that is learned during training as a normative prior over how the world should change when an action is correctly executed under a given goal. At inference time, this prior is used diagnostically: we detect mistakes by checking mismatches between the expected effect and the actually observed video evidence. This deliberate gap between how effect knowledge is learned (from correct executions) and how it is used (to diagnose erroneous ones) is specific to mistake detection and is not addressed by [5,6,7], which focus on forecasting normal effects or enabling control. We also highlight that such structured effect priors are naturally extensible to broader domains, including embodied AI and sequential decision-making systems.
>
> **References:**
>
> [1] Lee, Shih-Po, et al. "Error detection in egocentric procedural task videos." Proceedings of the IEEE/CVF Conference on Computer Vision and Pattern Recognition. 2024.
>
> [2] Huang, Wei-Jin, et al. "Modeling Multiple Normal Action Representations for Error Detection in Procedural Tasks." Proceedings of the Computer Vision and Pattern Recognition Conference. 2025.
>
> [3] OpenAI. GPT-5. OpenAI, 2025, <https://openai.com/gpt-5/>.
>
> [4] Google. Gemini 2.5 Pro. Google, 2025, <https://blog.google/technology/google-deepmind/gemini-model-thinking-updates-march-2025/>.
>
> [5] Dagan, Gautier, Frank Keller, and Alex Lascarides. "Learning the effects of physical actions in a multi-modal environment." arXiv preprint arXiv:2301.11845 (2023).
>
> [6] Souček, Tomáš, et al. "Look for the change: Learning object states and state-modifying actions from untrimmed web videos." Proceedings of the IEEE/CVF Conference on Computer Vision and Pattern Recognition. 2022.
>
> [7] Hu, Yucheng, et al. "Video prediction policy: A generalist robot policy with predictive visual representations." arXiv preprint arXiv:2412.14803 (2024).

---

> > ### Comment · Reviewer_JzSx · 2025-11-26
> >
> > Thanks for your detailed rebuttal, which has resolved most of my concerns. I have raised the rating from 2 to 6.

---

> > > ### Author Response · Authors · 2025-11-27
> > >
> > > Thank you for your careful review and positive update. We appreciate your time and thoughtful consideration of our work. We will upload the revised manuscript after completing discussions with all reviewers.

---

### Official Review · Reviewer_cXAQ · 2025-10-25

**Soundness:** 3
**Presentation:** 2
**Contribution:** 3
**Rating:** 6
**Confidence:** 4

**Summary:**

The manuscript presents Action Effect Modeling (AEM), a probabilistic framework that jointly captures the execution of an action and its resulting environmental state, arguing that many procedural errors are only detectable in the outcome rather than the motion itself. AEM first selects an effect‑rich frame using semantic relevance and visual quality cues, then fuses visual grounding with symbolic scene‑graph embeddings into a shared latent space to form an effect‑aware representation; a prompt‑based one‑class classifier is subsequently applied to detect mistakes. Evaluated on EgoPER and CaptainCook4D, AEM outperforms existing methods under the one‑class setting, demonstrating that integrating action outcomes yields more reliable error detection and provides a useful representation for downstream tasks.

**Strengths:**

* The paper identifies a genuine gap in current procedural‑mistake detection: most methods examine only how an action is performed, ignoring the _outcome_ that actually indicates an error.
* Explicitly modeling the effect of an action is a natural extension of procedural AI and can be transferred to domains such as industrial assembly, robotics, or medical procedure guidance.
* By jointly modeling execution and outcome through the proposed Action Effect Modeling (AEM) framework, the authors offer a conceptually novel solution that blends visual grounding, symbolic scene‑graph reasoning, and prompt‑based one‑class detection.
* The learnable effect token distills effect information into the action representation without requiring external LLMs at inference time, keeping the system lightweight.
* The prompt‑based detector aligns each action segment with a task‑specific textual prompt, explicitly modeling temporal dynamics and semantic context.
* AEM’s two‑branch multimodal supervision (visual + symbolic) is elegant and leverages large models to extract complementary cues.
* Reported AUC/EDA improvements  are convincing, and ablations support the claimed benefits of effect‑aware representations and cross‑modal alignment.
* Extensive ablation studies (effect‑frame sampling, state vs relation supervision, visual vs textual signals, cross‑modal alignment, dynamic fusion, prompt tuning) provide strong evidence for each design choice.

**Weaknesses:**

* In the overall training objective (Eq. (10) $L = L^{seg} + L^{eff} + L^{CL} + L^{det}$), the authors simply sum the four loss terms without any weighting or scaling factors. I am unsure whether the authors verified that these terms are in comparable magnitude; if not, a large loss could dominate the optimization and bias the learned representations. A brief ablation or sensitivity study showing the relative scales of the losses (or the inclusion of trainable weighting coefficients) would help the reader understand how the optimization balances execution modeling, effect learning, cross‑modal alignment, and detection.
* The manuscript does not provide a systematic hyper‑parameter study. None of the hyper‑parameters (learning rate, batch size, loss weights, temperature, etc.) are tuned or validated on a validation set. A brief ablation or sensitivity analysis, especially on the relative loss weights in Eq. (10) and on the temperature coefficient in the contrastive objectives, would strengthen confidence that the proposed method is robust and not overly sensitive to arbitrary hyper‑parameter choices.
* Heavy reliance on external large models for supervision is not quantified; downstream failure rates of the grounding or scene‑graph modules could propagate to the effect token but are not analyzed.
* No statistical significance tests or confidence intervals are reported, so the robustness of the reported gains is unclear.
* Evaluation is limited to two egocentric cooking datasets; generalization to other procedural domains remains untested, one of which is cited by the authors (Assembly101).
* Minor typographical and formatting inconsistencies—such as missing spaces and content overflowing in Table 2—reduce the document’s overall readability.

**Questions:**

Please refer to the section on weaknesses.

---

> ### Author Response · Authors · 2025-11-28
> **Response to Reviewer cXAQ (Part 1)**
>
> Thank you for your constructive feedback. We provide responses addressing each of your concerns in detail.
>
> **W1: Training objective.**
>
> |    Model    | Quesadilla AUC | Quesadilla EDA | Oatmeal AUC | Oatmeal EDA | Pinwheel AUC | Pinwheel EDA | Coffee AUC | Coffee EDA | Tea AUC | Tea EDA | All AUC | All EDA |
> |:-----------:|:--------------:|:--------------:|:-----------:|:-----------:|:------------:|:------------:|:----------:|:----------:|:-------:|:-------:|:-------:|:-------:|
> | Dyn weight  |      78.4      |      66.5      |    75.8     |    63.5     |     71.9     |     60.1     |    70.4    |    66.0    |  72.2   |  66.9   |  73.7   |  64.6   |
> | No weight   |      80.8      |      68.1      |    77.0     |    68.6     |     69.9     |     61.2     |    70.3    |    66.4    |  71.1   |  69.4   |  73.8   |  66.7   |
>
> *Table 1. Performance comparison across different subtasks on EgoPER dataset.*
>
> We experimented with several loss-weighting strategies but did not observe meaningful performance differences. In particular, we tried dynamically normalizing each loss term to match the magnitude of an anchor loss (e.g., the segmentation loss), and the results in the table above show that this scheme does not improve performance over the simple unweighted sum. By inspecting the loss curves on the validation set, we found that without any weighting, all four losses naturally converge to a comparable scale (approximately $0.01-0.1$), with the general ordering *seg loss* > *CL loss* > *eff loss* > *det loss*. These empirical findings indicate that no single loss dominates the optimization, suggesting that additional loss-weighting mechanisms are unnecessary in our model training.
>
> **W2: Hyper-parameter study.**
>
> | Metric | AMNAR | lr = 1e-4 | lr = 5e-4 | lr = 1e-3 | lr = 5e-3 |
> |:------:|:-----:|:---------:|:---------:|:---------:|:---------:|
> |  AUC   | 68.5  |   71.9    | **73.8**  |   74.1    |   70.4    |
> |  EDA   | 64.4  |   66.3    | **66.7**  |   62.8    |   65.2    |
>
> *Table 2. Comparison of the baseline and our model with different learning rates.*
>
> (a) **Learning rate:**  We experimented with multiple learning rates for model training, and a subset of the results is reported in the table above. A learning rate of $\text{lr} = 5\times10^{-5}$ yields the best performance when considering both AUC and EDA, while other settings still achieve better results than the baseline AMNAR [1]. This indicates that although the learning rate can slightly affect the detection performance, our method remains consistently better than the baseline.
>
> (b) **Relative loss weights:**  Please refer to our response to *W1 Training objective*, where we analyze different loss-weighting strategies and show that the simple unweighted sum in Eq. (10) is sufficient and does not lead to domination by any single term.
>
> (c) **Temperature coefficient:**  For the contrastive objectives, we follow the CLIP implementation [2] and use a learnable temperature parameter instead of a fixed hand-tuned value. This avoids additional hyper-parameter selection and has been empirically stable in our experiments.
>
> **W3: Reliance on external large models**
>
> | Model  | Quesadilla AUC | Quesadilla EDA | Oatmeal AUC | Oatmeal EDA | Pinwheel AUC | Pinwheel EDA | Coffee AUC | Coffee EDA | Tea AUC | Tea EDA | All AUC | All EDA |
> |:------:|:--------------:|:--------------:|:-----------:|:-----------:|:------------:|:------------:|:----------:|:----------:|:-------:|:-------:|:-------:|:-------:|
> | GPT-4o |    **80.8**    |      68.1      |    77.0     |    68.6     |     69.9     |     61.2     |    70.3    |    66.4    |  71.1   |  69.4   |  73.8   |  66.7   |
> | GPT-5  |      78.9      |    **69.1**    |  **79.1**   |  **70.1**   |   **70.4**   |     61.9     |  **70.4**  |  **66.4**  |  71.3   | **69.8**| **74.0**| **67.5**|
>
> *Table 3. Performance comparison across different subtasks on EgoPER dataset.*
>
> We quantify the supervision quality from external large models in two ways. First, we conduct a human study in Appendix B.2 to evaluate the generation by the large vision-language model (LVLM). Under strict manual assessment, the grounded objects, states, and relations in generated scene graphs achieve high accuracy (87%). Second, we explicitly examine how improved supervision quality affects mistake detection by replacing GPT-4o with a stronger LVLM (GPT-5) for scene graph generation. As shown in the table above, this leads to higher mistake detection performance. These results indicate that our framework is robust to supervision noise and can further benefit from advances in modern LVLMs.

---

> ### Author Response · Authors · 2025-11-28
> **Response to Reviewer cXAQ (Part 2)**
>
> **W4: Statistical significance tests and confidence intervals**
>
> | Metric | AMNAR | Run #1 | Run #2 | Run #3 | Run #4 | Run #5 | Mean ± 95% CI |
> |:------:|:-----:|:------:|:------:|:------:|:------:|:------:|:-------------:|
> |  AUC   | 68.50 | 73.74  | 74.14  | 74.59  | 74.22  | 73.80  |  74.10 ± 0.42 |
> |  EDA   | 64.40 | 64.63  | 66.53  | 66.37  | 67.01  | 66.70  |  66.25 ± 1.16 |
>
> *Table 4. Comparison with baseline across five runs.*
>
> We run our model five times and report the mean and 95% confidence intervals for both AUC and EDA (see table above). The results show small variance across runs and consistent improvements over the baseline AMNAR [1], indicating that the reported gains are robust rather than due to random fluctuations.
>
> **W5: Evaluation on dataset in other domains.**
>
> |  Method  |  AUC  |  EDA  |
> |:--------:|:-----:|:-----:|
> |  Random  | 51.2  | 49.7  |
> |  EgoPER  | 55.2  | 66.2  |
> |  AMNAR   | _56.5_ | _69.9_ |
> | **Ours** | **57.3** | **70.7** |
>
> *Table 5. Performance comparison of different methods on HoloAssist dataset.*
>
> We follow AMNAR [1] to benchmark our model on the HoloAssist dataset [3] focusing on the assembly scenario. The results in the table above show that our proposed model outperforms.
>
> **W6: Typographical and formatting inconsistencies.**
>
> We will fix the noted typographical issues and adjust Table 2 to avoid content overflow, and will ensure consistent formatting in the revised manuscript.
>
> **References:**
>
> [1] Huang, Wei-Jin, et al. "Modeling Multiple Normal Action Representations for Error Detection in Procedural Tasks." Proceedings of the Computer Vision and Pattern Recognition Conference. 2025.
>
> [2] Radford, Alec, et al. "Learning transferable visual models from natural language supervision." International Conference on Machine Learning. PMLR, 2021.
>
> [3] Wang, Xin, et al. "HoloAssist: an egocentric human interaction dataset for interactive AI assistants in the real world." Proceedings of the IEEE/CVF International Conference on Computer Vision. 2023.

---

### Official Review · Reviewer_pw23 · 2025-11-02

**Soundness:** 3
**Presentation:** 2
**Contribution:** 4
**Rating:** 6
**Confidence:** 4

**Summary:**

The paper studied procedural mistake detection and incorporates the effect of actions alongside their execution, which seems to be original. To do this, it introduces Action Effect Modeling (AEM), a probabilistic framework that jointly models action execution, and the resulting outcomes (the hardest part of the paper to understand), and automatic frame selection. Evaluations on egocentric video benchmarks demonstrate that AEM outperforms prior state-of-the-art methods in both action and outcome-based mistake recognition. The ablation study is thorough, assessing multiple important modeling questions.

**Strengths:**

- S1 The main methodological contribution that incorporates both the action execution and its effect into a probabilistic framework is original in the procedural mistake detection space.

- S2 The paper builds on the growing literature in procedural mistake detection, enabling measured comparisons by using common pipeline steps from recent works.

- S3 The paper includes two contemporary and popular procedural mistake detection benchmarks in its evaluation.  Generally the proposed work achieves quick strong improvements over the state of the art.  The experiments also include ablation analysis over many key factors in the modeling, demosntrating robust performance.

- S4 The effect frame sampling is very interesting.  It's a shame the paper had to rush through its discussion.  It seems that this notion of "bayesian" frame selection may have a broader potential in problems like this.

**Weaknesses:**

- W1  The prompt based detector is not well explained, not does it seem to be properly analyzed in the results.  Yet, it is claimed to be a primary contribution.  Is this alignment not needed in general for procedural mistake detection methods?  This part of the paper is very unclear.  And, considering its importance in the overall paper, this significanlty detracts from the quality of the paper.

- W2 The paper seems heavily dependent on GPT4o for numerous functionality.  Notwithstanding the fact that GPT4o is a closed model and hence hard to directly analyze, one of the key points of the paper is the relational graph that is

- W3 The paper is wrought with exposition terseness, small mistakes, and nonuniform level of description across various parts, making it harder to follow than necessary.
  - W3a `action effects` seems to be defined in various ways throughout the paper, always implicitly.  It would be very helpful to more clearly define action effect.  Furthermore, how many action effects are studied?  Is it just the two that are ablated?  Do each of these have a set of features associated with them?  This part of the paper is far from declarative and seriously distracts from the contributions.
  - W3b Isn't the first term in equation 1 $P(\hat{y}|\mathbf{X},e_i,f_e)$?  Any notion of independent of the effect frame is not obvious.
  - W3c The paper has some unfortunate grammatical errors.  EG L077 "from vision-language model" ?  from a? models?  Typographical errors.  EG L430 "clarityfurther". And, there are also some seemingly incorrect sentence structures.  L126 "However, mistake detection in procedural videos is more goal-oriented" ... mistake-detection is more goal oriented or the problem of procedural video modeling is more goal oriented?

- W4 Given the dependence on numerous pretrained (and sometimes closed) models, it is unfortunate the paper does not analyze the generalizability and robustness to alternatives.  This weakens the impact of the work.

**Questions:**

- Q1 What text-visual embedding model is used for the effect frame sampling?
- Q2 How are the models actually learned?  The complexity of the marginal and sampling across frames of the video seems like the joint learning (which seems to be proposed) is very "big."  What is happening here really?
- Q3 Until section 4.4 it is not clear that the action segmentation method is also learned (or fine-tuned) as part of the process.  This is unfortunate, since the action segmentation does seem to impact the results.  One wonders how the other methods would perform using these segments (i.e., is the gain from better segments or is it from action effect modeling).

---

> ### Author Response · Authors · 2025-11-27
> **Response to Reviewer pw23 (Part 1)**
>
> Thank you for your constructive feedback. We provide responses addressing each of your concerns in detail.
>
> **W1: Explanation and results analysis for prompt-based detector**
>
> **1) Explanation:**  Prior methods detect procedural mistakes through frame-level classification and largely ignore the temporal and semantic structure of each action segment. In contrast, our prompt-based detector uses text prompts to guide the aggregation of temporally structured object-centric information. Such alignment between action segments and task-aware prompts is a generally useful principle for mistake detection, which is overlooked in existing approaches.
>
> **2) Result analysis.**
>
> | Method        | AUC  | EDA  |
> |--------------|------|------|
> | EgoPER       | 62.0 | 57.0 |
> | AMNAR        | **68.5** | _64.4_ |
> | Ours w/o AEM | _67.6_ | **65.6** |
>
> *Table: Comparison with prior methods on the EgoPER dataset.*
>
> We remove the Action Effect Modeling (AEM) module and keep only the segmentation backbone plus our mistake detector to quantify the effect of prompt-based mistake detector. As the table above shows, this simplified variant already achieves performance comparable to or even exceeding the latest mistake detection methods [1,2] with the same segmentation backbone but more complicated designs, highlighting the standalone effectiveness of the detector.
>
> **W2: Reliance on close-source GPT-4o model**
>
> | Model     | Quesadilla AUC | Quesadilla EDA | Oatmeal AUC | Oatmeal EDA | Pinwheel AUC | Pinwheel EDA | Coffee AUC | Coffee EDA | Tea AUC | Tea EDA | All AUC | All EDA |
> |-----------|----------------|----------------|-------------|-------------|--------------|--------------|------------|------------|---------|---------|---------|---------|
> | Qwen3-VL  | 77.8 | 69.0 | 77.3 | 68.8 | 70.0 | **62.2** | 70.2 | 64.3 | **71.4** | 68.7 | 73.3 | 66.6 |
> | GPT-4o    | **80.8** | 68.1 | 77.0 | 68.6 | 69.9 | 61.2 | 70.3 | 66.4 | 71.1 | 69.4 | 73.8 | 66.7 |
> | GPT-5     | 78.9 | **69.1** | **79.1** | **70.1** | **70.4** | 61.9 | **70.4** | **66.4** | 71.3 | **69.8** | **74.0** | **67.5** |
>
> *Table: Performance comparison across different subtasks on the EgoPER dataset.*
>
> Our framework does not rely on any specific large model such as GPT-4o [3]. Instead, any large vision-language model (LVLM) can be integrated into it for action scene graph generation. In the table above, we replace GPT-4o with some alternative models for scene graph generation, including the open-source Qwen3-VL (30B) model [4]. The results illustrate that the Qwen3-VL variant achieves comparable performance to the GPT-4o variant, showing that the graph construction is robust to the specific choice of LVLM.
>
> **W3a: Definition of Action Effects and associated features**
>
> **1) Definition:**  In this paper, we define Action Effect as the outcome reflected on the manipulated objects after an action, for example changes in their state or spatial arrangement.
>
> **2) Effect features:**  We study two types of effects in our benchmarks: object state effects and spatial relation effects, since together they cover most observable outcomes in the dataset.  Each type of action effects is associated with a dedicated effect token aligned with corresponding supervision signal during training, as described in Section 3.4 (Lines 264-273) and illustrated in Figure 3. Our framework is flexible and not limited to these two effects and can be extended to finer grained dimensions (e.g., color, shape, pose) or higher level properties (e.g., quantity and temperature) by adding additional effect tokens.
>
> **W3b: Explanation for Eq. (1)**
>
> Since $e_i$ is obtained conditioned on $f_e$ through $P(e_i \mid f_e, \mathbf{X})$, we assume that once the effect descriptor $e_i$ is known, the specific frame $f_e$ no longer provides additional information for mistake prediction, i.e.,
> $$
> \hat y \perp f_e \mid (\mathbf X, e_i) \rightarrow P(\hat{y}\mid e_i, f_e, \mathbf{X}) = P(\hat{y}\mid e_i, \mathbf{X}).
> $$
> We will make this conditional independence assumption explicit in the revision:
>
> *(Line 159) "Here we assume that once the segment feature $\mathbf{X}$ and the effect descriptor $e_i$ are obtained, the specific effect frame $f_e$ does not provide additional information for mistake prediction, i.e., $\hat y \perp f_e \mid (\mathbf X, e_i)$, thus $P(\hat{y}\mid \mathbf{X}, e_i, f_e) = P(\hat{y}\mid \mathbf{X}, e_i)$."*
>
> **W3c: Grammatical errors**
>
> We will correct the grammatical and typographical issues mentioned in the review as follows.
>
> - Line 077 will be revised from *“from vision-language model”* to *“from vision–language models”*.
>
> - Line 430 will be corrected from *“clarityfurther”* to *“clarity further”*.
>
> - Line 126 will be rewritten to state that *“mistake detection in procedural videos is inherently goal oriented"*, to avoid ambiguity about whether we refer to the mistake detection problem or the video modeling task.

---

> ### Author Response · Authors · 2025-11-27
> **Response to Reviewer pw23 (Part 2)**
>
> **W4: Generalizability and robustness to alternative models**
>
> Please refer to our response for "W2: Reliance on close-source GPT-4o model.", where we compare the performance to alternative models including an open-source one.
>
> **Q1: Text–visual embedding model**
>
> We use the off-the-shelf EVA-02 CLIP encoders [5] for both image and text embedding, as described in Section 4.1 (Lines 352–353).
>
> **Q2: Model learning and computational efficiency**
>
> | Model | Training speed (sec/step) ↓ | Inference speed (sec/video) ↓ |
> |:-----:|:---------------------------:|:------------------------------:|
> | AMNAR | 0.94                        | 0.29                           |
> | Ours  | 0.28                        | 0.12                           |
>
> *Table: Training and inference efficiency comparison.*
>
> Eq. (1) is a probabilistic formulation that conceptually corresponds to the modules in our framework, presenting the learning process from a mathematical perspective.
>
> Besides, the overall complexity of our model remains limited. The frame sampling component is executed offline during data preprocessing and does not introduce additional training cost. During training, the action segmentation backbone, the effect-aware learning module, and the mistake detection module are jointly optimized in an end-to-end manner. To further demonstrate efficiency, we report the average training time per step and the average inference time per video compared with a prior approach [2], and the timing results in the table above indicate that the overall computational overhead is limited.
>
> **Q3: Action segmentation method**
>
> | Action Seg | Quesadilla AUC | Quesadilla EDA | Oatmeal AUC | Oatmeal EDA | Pinwheel AUC | Pinwheel EDA | Coffee AUC | Coffee EDA | Tea AUC | Tea EDA | Avg AUC | Avg EDA |
> |------------|----------------|----------------|-------------|-------------|--------------|--------------|------------|------------|---------|---------|---------|---------|
> | Pred       | 80.8 | 68.1 | 77.0 | 68.6 | 69.9 | 61.2 | 70.3 | 66.4 | 71.1 | 69.4 | 73.8 | 66.7 |
> | GT         | 92.8 | 78.4 | 94.2 | 75.0 | 80.7 | 72.0 | 80.8 | 71.3 | 84.8 | 76.7 | 86.7 | 74.7 |
>
> *Table: Effect of ground-truth versus predicted action segmentation.*
>
> We clarify that the action segmentation is learned within our joint end-to-end training framework, as specified in Section 3.6 (Lines 318–323). To quantify its impact on mistake detection, we evaluate our method by replacing predicted segments with ground-truth segments (results shown in the table above). Using ground-truth segmentation leads to a significant improvement in mistake detection performance, indicating that better segmentation should consistently improve detection performance.
>
> **References**
> [1] Lee, Shih-Po, et al. "Error detection in egocentric procedural task videos." Proceedings of the IEEE/CVF Conference on Computer Vision and Pattern Recognition. 2024.
> [2] Huang, Wei-Jin, et al. "Modeling Multiple Normal Action Representations for Error Detection in Procedural Tasks." Proceedings of the Computer Vision and Pattern Recognition Conference. 2025.
> [3] Hurst, Aaron, et al. “GPT-4o System Card.” arXiv preprint arXiv:2410.21276, 2024.
> [4] Yang, An, et al. "Qwen3 technical report." arXiv preprint arXiv:2505.09388 (2025).
> [5] Fang, Yuxin, et al. "Eva-02: A visual representation for neon genesis." Image and Vision Computing 149 (2024): 105171.

---

### Official Review · Reviewer_nqhL · 2025-11-03

**Soundness:** 3
**Presentation:** 3
**Contribution:** 2
**Rating:** 4
**Confidence:** 4

**Summary:**

This work proposes AEM for procedural mistake detection, modeling both action execution and effects through effect frame selection, multimodal features (visual and symbolic), and a contrastive prompt detector. It achieves better on EgoPER and CaptainCook4D under OCC.

**Strengths:**

Strong empirical results on two benchmarks, with ablations validating key components.

**Weaknesses:**

1. Novelty is limited, as the integration of scene graphs and VLMs builds on existing work (e.g., Hurst et al., 2024) and offers little beyond fusion.
2. Benchmarks are narrow (only two datasets) and lack evaluation across diverse domains, such as assembly or medical procedures, as mentioned in the introduction.
3. The prompt-based detector seems straightforward, and gains over baselines like ProtoMD are modest in some metrics.
4. The methodology sections are dense, with equations (e.g., Eq. 1) that need more intuitive explanations.
5. Some sections (e.g., the methodology) could benefit from additional pseudocode or flowcharts to improve clarity.

**Questions:**

1. How does AEM handle actions with delayed or invisible effects (e.g., internal changes in objects)? Could AEM be extended to online detection without full segment access?
2. Could you compare computational overhead to simpler execution-only methods?
3. Why only two effect descriptors (states and relations)? Are there plans to extend to more?
4. How robust is the effect of frame sampling to noisy or occluded videos?
5. What is the impact of VLM choice (e.g., alternatives to Hurst et al., 2024) on performance?

---

> ### Author Response · Authors · 2025-11-26
> **Response to Reviewer nqhL (Part 1)**
>
> Thank you for your constructive feedback. We provide responses addressing each of your concerns in detail.
>
> **W1: Limited novelty by integration of scene graphs and VLMs.**
>
> Our core novelty lies in proposing that action-effect modeling is central to mistake detection. The model learns what each action is expected to do to task relevant objects, and then uses this learned effect prior to decide whether an observed execution is correct or erroneous. This perspective has not been considered by prior mistake detection work [1,2] which primarily operates at the action level.
>
> Our modeling of the action effect is also innovative. We use a VLM to build an object centric scene graph, then turn object states and spatial relations into a structured action effect representation that encodes expected object changes under correct execution, and is applied diagnostically at inference. It goes beyond a simple integration, but a way to achieve object-centric action effect modeling for mistake detection.
>
> **W2: Benchmark in other domains.**
>
> | Method   |  AUC  |  EDA  |
> |:--------:|:-----:|:-----:|
> | Random   | 51.2  | 49.7  |
> | EgoPER   | 55.2  | 66.2  |
> | AMNAR    | _56.5_ | _69.9_ |
> | **Ours** | **57.3** | **70.7** |
>
> *Table: Performance comparison of different methods on HoloAssist dataset.*
>
> We follow AMNAR [2] to benchmark our model on the HoloAssist dataset [3] focusing on the assembly scenario. The results in the table above show that our proposed model outperforms.
>
> **W3: Straightforward prompt-based detector, and modest gains over baselines ProtoMD.**
>
> We want to point out that our baseline **does not include ProtoMD**. We argue that the proposed prompt-based detector is both effective and conceptually meaningful. Previous work [1,2] relies on frame features which largely ignores the temporal structure, while our method uses learnable prompts to aggregate temporally organized information, enabling more accurate mistake detection.
>
> **W4: Dense methodology sections, equations (e.g., Eq. 1) need intuitive explanations.**
>
> We provide a more intuitive explanation for Eq. (1) below which will be added to the revised paper:
>
> *from Line 159: “… Intuitively, Eq. (1) indicates that we consider every possible effect frame and each of its associated action-effect descriptors, estimate the likelihood of each (f_e, e_i) combination, evaluate whether it suggests a mistake, and then aggregate these weighted probabilities to obtain the final prediction.”*
>
> **W5: Additional pseudocode or flowcharts for improved clarity.**
>
> The pseudocode for action scene graph construction and decomposition is provided in Appendix A.3 due to the page limit. In the revised version, we will add a clearer reference in the main paper.
>
> **Q1: The ways AEM handle delayed or invisible effects, and to be extended to online detection.**
>
> Our current model primarily targets actions whose effects yield at least partially observable visual cues in RGB frames. For actions with strongly delayed or purely internal effects, the computer vision models, including ours, would be limited unless there is access to additional modalities (e.g., specialized sensors) or stronger domain priors.
>
> Regarding the extension to online detection, a potential idea can be employed by maintaining provisional effect tokens from partial segments within a causal temporal window and incrementally updating them as new frames arrive.
>
> **Q2: Comparison of computational overhead to execution-only methods.**
>
> | Model        | Training speed (sec/step) ↓ | Inference speed (sec/video) ↓ |
> |:------------:|:---------------------------:|:------------------------------:|
> | AMNAR        | 0.94                        | 0.29                           |
> | Ours w/o AEM | 0.21                        | 0.10                           |
> | Ours w/ AEM  | 0.28                        | 0.12                           |
>
> *Table: Training and inference efficiency comparison.*
>
> We report both the average training time per step and the average inference time per video, comparing our models with Action-effect Modeling (AEM), our execution-only model without AEM, and an execution-only previous approach [2]. The results in the table above show that both variants of our model are efficient.

---

> > ### Author Response · Authors · 2025-11-26
> > **Response to Reviewer nqhL (Part 2)**
> >
> > **Q3: Extension of effect descriptors.**
> >
> > We chose object state and spatial relation as effect descriptors because these two cues account for the majority of observable action outcomes and procedural mistakes in our benchmarks. However, AEM is not restricted to this design. The framework is extensible as effect descriptors can be further decomposed into finer-grained dimensions (e.g., color, shape, pose, spatial extent) or higher-level properties (e.g., quantity, temperature), which can be incorporated by introducing additional effect tokens that model each perspective.
> >
> > **Q4: How robust is the effect of frame sampling to noisy or occluded videos?**
> >
> > | Method     |  AUC  |  EDA  |
> > |:----------:|:-----:|:-----:|
> > | w/o Effect | 67.6  | 65.6  |
> > | Last Frame | _70.6_ | _65.7_ |
> > | **Ours**   | **73.8** | **66.7** |
> >
> > *Table: Comparison of frame selection strategies.*
> >
> > Our effect-frame sampling remains robust to moderate noise and short occlusions because it scores all candidate frames using both semantic relevance and visual quality. As shown in the table above (Table 3(b) in the main paper), the last-frame baseline often suffers from poor visual quality or irrelevant content, leading to weaker performance, whereas our dynamic sampling provides consistently stronger results.
> >
> > For the severe or prolonged occlusions which is challenging for all visual models including ours, some potential solutions may help to mitigate, such as leveraging multi-view visual inputs.
> >
> > **Q5: What is the impact of VLM choice (e.g., alternatives to Hurst et al., 2024) on performance?**
> >
> > | Model      | Quesadilla AUC | Quesadilla EDA | Oatmeal AUC | Oatmeal EDA | Pinwheel AUC | Pinwheel EDA | Coffee AUC | Coffee EDA | Tea AUC | Tea EDA | All AUC | All EDA |
> > |:----------:|:--------------:|:--------------:|:-----------:|:-----------:|:------------:|:------------:|:----------:|:----------:|:-------:|:-------:|:-------:|:-------:|
> > | Qwen3-VL   | 77.8           | 69.0           | 77.3        | 68.8        | 70.0         | **62.2**     | 70.2       | 64.3       | **71.4** | 68.7    | 73.3    | 66.6    |
> > | GPT-4o     | **80.8**       | 68.1           | 77.0        | 68.6        | 69.9         | 61.2         | 70.3       | 66.4       | 71.1    | 69.4    | 73.8    | 66.7    |
> > | GPT-5      | 78.9           | **69.1**       | **79.1**    | **70.1**    | **70.4**     | 61.9         | **70.4**   | **66.4**   | 71.3    | **69.8** | **74.0** | **67.5** |
> >
> > *Table: Performance comparison across different subtasks on EgoPER dataset.*
> >
> > Our method continues to benefit from advances in underlying VLMs, and open-source models can serve as a cost-effective alternative for generating scene graphs while maintaining strong performance. To verify this, in addition to the GPT-4o [4] model used in the paper, we evaluated variants using the latest closed-source GPT-5 [5] and the open-source Qwen3-VL (30B) [6]. As shown in the table, the GPT-5 variant achieves the best mistake-detection performance, surpassing GPT-4o, while Qwen3-VL performs comparably to GPT-4o.
> >
> > **References:**
> >
> > [1] Lee, Shih-Po, et al. "Error detection in egocentric procedural task videos." Proceedings of the IEEE/CVF Conference on Computer Vision and Pattern Recognition. 2024.
> >
> > [2] Huang, Wei-Jin, et al. "Modeling Multiple Normal Action Representations for Error Detection in Procedural Tasks." Proceedings of the Computer Vision and Pattern Recognition Conference. 2025.
> >
> > [3] Wang, Xin, et al. "Holoassist: an egocentric human interaction dataset for interactive AI assistants in the real world." Proceedings of the IEEE/CVF International Conference on Computer Vision. 2023.
> >
> > [4] Hurst, Aaron, et al. "GPT-4o System Card." arXiv preprint arXiv:2410.21276, 2024.
> >
> > [5] OpenAI. GPT-5. OpenAI, 2025, <https://openai.com/gpt-5/>.
> >
> > [6] Yang, An, et al. "Qwen3 technical report." arXiv preprint arXiv:2505.09388 (2025).

---

### Author Response · Authors · 2025-12-02
**Summary of Reviewer Feedback and Rebuttal Progress**

We thank the AC and all reviewers for their time, efforts, and constructive discussions. Below we summarize the reviews and the main progress during the rebuttal phase.

---

### 1. Scores and Confidence

- Reviewer nqhL: Rating **4**, Confidence **4**
- Reviewer pw23: Rating **6**, Confidence **4**
- Reviewer cXAQ: Rating **6**, Confidence **4**
- Reviewer JzSx: Rating improved from **2 → 6**, Confidence **3**

---

### 2. Main Advantages Highlighted by Reviewers

- Novel action-effect probabilistic framework (pw23, cXAQ): Recognized as an original formulation for procedural mistake detection.

- Strong empirical results with thorough ablations (nqhL, pw23, cXAQ, JzSx): Experiments and ablation studies validate AEM’s components and provide general insights beyond this specific task.

- Effect-aware representation design (effect frames and effect token) (pw23, cXAQ, JzSx): Effect-frame sampling and learnable effect tokens are viewed as interesting and broadly applicable.

- Elegant multimodal supervision and use of large models ( cXAQ, JzSx): Two-branch visual + symbolic supervision is considered well designed and effective.

---

### 3. Major Concerns and Our Responses

a) Dataset Diversity and Generalization (nqhL, cXAQ)

- **Concern:** Evaluation originally used only two cooking datasets, raising questions about domain generalization.
- **Response:** We added experiments on HoloAssist (assembly tasks). AEM achieves 57.3% AUC and 70.7% EDA, outperforming random, EgoPER, and AMNAR baselines, showing that AEM generalizes beyond cooking domains.

b) Dependence on Vision-Language Models (pw23, cXAQ)

- **Concern:** The framework might rely too heavily on GPT-4o for scene-graph generation and might not be robust to the choice of LVLM.
- **Response:** We replaced GPT-4o with Qwen3-VL and GPT-5. On EgoPER, Qwen3-VL achieves 73.3% / 66.6% AUC/EDA and GPT-5 achieves 74.0% / 67.5%, comparable to or better than GPT-4o (73.8% / 66.7%). This shows that our framework is robust to the LVLM choice and can effectively leverage open-source models.

c) Runtime and Modular Design (nqhL, pw23, JzSx)

- **Concern:** The modular structure might be computationally heavy, and it was unclear whether segmentation is learned end-to-end.
- **Response:** All components are trained end-to-end, with frame sampling performed offline. In practice, AEM is faster than AMNAR:
  - Training: 0.28 s vs 0.94 s per step
  - Inference: 0.12 s vs 0.29 s per video
  Using ground-truth segments further boosts mean AUC from 73.8% to 86.7%, indicating that better segmentation quality can further improve detection.

d) Prompt-Based Detector and Model Components (nqhL, pw23)

- **Concern:** The prompt-based detector appears simple, shows modest gains, and was not clearly explained or analyzed.
- **Response:** We clarified that the detector aligns each action segment with a task-specific textual prompt, guiding aggregation of temporally structured object-centric features. This introduces temporal and semantic context that frame-level classifiers lack. A variant using only the segmentation backbone plus our detector (without AEM) still reaches 67.6% AUC and 65.6% EDA, confirming that the detector itself is effective.

---

We hope this summary provides a clear and concise overview of the progress made during the rebuttal phase and how the reviewers’ concerns were addressed. We appreciate the AC’s time and consideration, and we remain committed to further improving the clarity and impact of our work in the revision.

---

### Meta-Review · Area_Chair_tCRK · 2026-01-06

**Summary:**

The reviewers raised several concerns regarding novelty, limited experiments (only two datasets), lack of robustness to other large multimodal models and lack of clarity about Prompt-Based detector. The rebuttal addressed almost all the concerns. However, the AC believes that the the concern regarding limited novelty (i.e., the proposed method is the integration of scene graphs and VLM) is not fully addressed by the rebuttal. However, considering several other points such as extensive experiments and interesting design as mentioned by other reviewers, the AC recommends acceptance.

**Reviewer Concerns:**

I believe all concerns have been addressed. Specifically, the authors have included new results on an additional dataset, evaluated performance using other large multimodal models to demonstrate robustness, and clarified the Prompt-Based Detector along with its effectiveness.

**Reviewer Scores:**

The lowest score reviewer has already changed their score to 6 (originally 2). The other reviewers are positive about this work and I believe they would give at least 6.

---

### Decision · Program_Chairs · 2026-01-26

Accept (Poster)